# p140Cap inhibits β-Catenin in the breast cancer stem cell compartment instructing a protective anti-tumor immune response

The p140Cap adaptor protein is a tumor suppressor in breast cancer associated with a favorable prognosis. Here we highlight a function of p140Cap in orchestrating local and systemic tumor-extrinsic events that eventually result in inhibition of the polymorphonuclear myeloid-derived suppressor cell function in creating an immunosuppressive tumor-promoting environment in the primary tumor, and premetastatic niches at distant sites. Integrative transcriptomic and preclinical studies unravel that p140Cap controls an epistatic axis where, through the upstream inhibition of β-Catenin, it restricts tumorigenicity and self-renewal of tumor-initiating cells limiting the release of the inflammatory cytokine G-CSF, required for polymorphonuclear myeloid-derived suppressor cells to exert their local and systemic tumor conducive function. Mechanistically, p140Cap inhibition of β-Catenin depends on its ability to localize in and stabilize the β-Catenin destruction complex, promoting enhanced β-Catenin inactivation. Clinical studies in women show that low p140Cap expression correlates with reduced presence of tumor-infiltrating lymphocytes and more aggressive tumor types in a large cohort of real-life female breast cancer patients, highlighting the potential of p140Cap as a biomarker for therapeutic intervention targeting the β-Catenin/ Tumor-initiating cells /G-CSF/ polymorphonuclear myeloid-derived suppressor cell axis to restore an efficient anti-tumor immune response.

Breast cancer (BC) is one of the most commonly cancers in USA with 290.560 estimated new cases and 43.780 estimated deaths in 2022[1,2]. Due to the heterogeneous clinical behavior of BC in response to currently available multimodal treatments[3], the development of new targeted therapies and biomarkers for improved patient management remains an unmet clinical need[4]. The existence of a subpopulation of tumor-initiating cells (TIC), able to resist therapies and drive disease progression and metastasis, is a major underlying cause of BC heterogeneity[5,6]. Accordingly, the intrinsic TIC content can inform the biological and clinical heterogeneity of BCs, and selective targeting of the mechanisms underlying uncontrolled TIC expansion/dissemination can efficiently curb tumorigenesis and metastatic progression[5,6]. Therefore, a deeper understanding of the mechanisms controlling the

TIC compartment in BC holds great promise for the development of new therapeutic strategies. Both intrinsic (tumor cell-autonomous) and extrinsic (depending on the tumor microenvironment, TME) mechanisms of cell fate determination have been defined, whose alteration can drive the appearance and expansion of cells with stemness traits in the bulk tumor population in BC[5,7,8]. The aberrant regulation of the Wnt/β-Catenin signaling pathway, leading to unchecked cytoplasmic stabilization and intra-nuclear translocation of β-Catenin, with increased downstream transcription of Wnt target genes, has emerged as a major cell-autonomous mechanism involved in the acquisition of stemness traits in different cancer types[9–13]. However, despite extant correlative evidence between aberrant β-Catenin activity and clinico-biological aggressiveness reported in

✉ e-mail: salvatore.pece@ieo.it; paola.defilippi@unito.it

different BC studies[14–16], the actual pathogenetic relevance of the Wnt/β-Catenin signaling to breast tumorigenesis remains elusive. This largely depends by the absence of well-defined genetic mutations of the Wnt/β-Catenin pathway in BC[17], at variance with other types of cancer, such as colon cancer. Recent evidence has highlighted a key role for the Wnt/β-Catenin pathway in the bidirectional cross talk between tumor cells and TME cells[18,19], where this pathway appears to promote the formation of a TME conducive to tumor progression and metastasis, inducing tumor-promoting modifications of the immune infiltrate[20,21]. Whether the dysfunction of tumor suppressor pathways relevant to BC can hijack the Wnt/β-Catenin signaling pathway to instruct a tumor-promoting immune response in the TME has been poorly investigated[17,22].

One such mechanism of tumor suppression in BC is represented by the p140Cap adaptor protein, encoded by the *SRCIN1* gene, whose expression is associated with a significantly reduced probability of developing distant recurrence and improved overall survival, in particular in HER2-positive BC patients[23,24]. The tumor suppressor activity of p140Cap has been largely attributed to its intrinsic ability to interact with proteins involved in different cancer-associated biological networks[25]. Notably, among its protein-protein interactions, p140Cap has also been described as a direct binder of β-Catenin through two coiled-coil regions in its 351-1051 amino-acid portion[26], although the functional significance of this interaction and its possible relevance to BC remain elusive.

In this study, we present evidence that, by its ability to interact with and inhibit β-Catenin, p140Cap can orchestrate an anti-tumor immune response, influencing the composition of the TME immune infiltrate to prevent the establishment of a tumor conducive immune environment. Indeed, we show that p140Cap can efficiently counteract the mobilization and intratumor accumulation of polymorphonuclear myeloid-derived suppressive cells (PMN-MDSCs), which create a permissive environment favoring tumor progression and metastatic spreading. Mechanistically, we establish that this p140Cap function relies on its ability to inhibit the Wnt/β-Catenin pathway by entering in and stabilizing the β-Catenin destruction machinery, thereby promoting β-Catenin inactivation. This results in restriction of the pool of TICs and, in turn, of secretion of GCS-F, an inflammatory cytokine involved in mobilization and tumor infiltration of PMN-MDSCs that restrict the infiltration of anti-tumor immune cells such as Natural Killer, M1-macrophages and T-lymphocytes. Establishing the clinical relevance of these findings, the retrospective analysis of a large consecutive cohort of women with BC reveals that tumors with high p140Cap expression show significantly higher stromal accumulation of tumor-infiltrating lymphocytes (TILs). Collectively, our findings highlight the existence of an epistatic β-Catenin/TIC/G-CSF/PMN-MDSCs axis, arguing for its potential actionability for therapeutic intervention to restore an efficient anti-tumor immune response.

## Results

### p140Cap expression correlates with an increased presence of tumor-infiltrating lymphocytes (TILs) and influences inflammatory responses in human breast cancers

Given the increasingly recognized relationship between immune TME and BC disease outcome[27,28], we asked whether p140Cap expression might affect the establishment of an efficient anti-tumor immune response at the level of the TME infiltrate. The presence of TILs in the TME is indicative of a proficient anti-tumor immune activity and represents a robust independent prognostic biomarker in BC[29,30]. We analyzed the correlation between p140Cap expression and stromal TILs density using hematoxylin and eosin (H&E)-fixed samples from a retrospective consecutive cohort of 622 female patients, with clinicopathological follow-up (IEO cohort, see Methods). Patients were classified according to their p140Cap expression based on an intensity score from 0 to 3 (0–0.5+, negative; 1+, weak positive; 2+, moderate

positive; and 3+, strong positive), and stratified as p140Cap^LOW (<1) or p140Cap^HIGH (≥1), as previously described[23]. TILs density was assessed by detecting the total quantity of mononuclear immune cells as a function of the overall percentage of the stromal area within the borders of the invasive tumor (see also Methods and international consensus guidelines[31]). Data on TILs were available only for 390 patients due to the excessive compactness of tumor nests, with not quantifiable stromal areas, and/or absence of tumor borders in the remaining cases. This analysis revealed a significant correlation between p140Cap^HIGH status and enhanced TILs infiltration in the entire population (OR = 2.25, CI 95% 1.13–4.48, p = 0.02) and in HER2⁻ patients (OR = 2.23, CI 95% 1.01–4.91, p = 0.04), with a similar trend in the subgroup of HER2⁺ patients (OR = 2.13, CI 95% 0.50–9.03, P = 0.30) (Fig. 1a, b), albeit not statistically significant likely due to the very low number of cases. Consistent with the association between stromal TILs and more favorable prognosis in BC[29,30], we found a significant univariate correlation between presence of stromal TILs and better overall survival in our BC patient cohort (HR = 2.35, CI 95% 1.19–4.30, p = 0.009) in all patients and in HER2⁺ patients (HR = 3.12, CI 95% 0.97–8.79, p = 0.038), with a consistent behavior, albeit not statistically significant, in the subpopulations of HER2⁻ patients (HR = 1.89, CI 95% 0.77–4.02, p = 0.13) (Supplementary Fig. 1). Phenotypic characterization of the immune infiltrate of p140Cap^HIGH tumors showed concomitant presence of both CD3⁺/T-lymphocytes and CD20⁺/B-lymphocytes, indicative of a proficient TME immune response (Supplementary Fig. 2). In keeping with these findings, Gene Set Enrichment Analysis (GSEA[32,33]) of the transcriptomic profiles of a cohort of 1095 BC patients from The Cancer Genome Atlas (TCGA)[34], stratified by transcript levels of the *SRCIN1* gene, revealed that the HALLMARK_IN-FLAMMATORY_RESPONSE gene set is one of the top-ranking signatures inversely associated with the *SRCIN1* transcript levels (Fig. 1c). In a subgroup analysis of the patients of this TGCA cohort, stratified for the different BC molecular subtypes (i.e. luminal, HER2⁺ and triple-negative-TNBC), we found that, compared to luminal and HER2⁺ patients, TNBC patients showed significantly reduced expression levels of *SRCIN1* transcript and p140Cap protein levels (Supplementary Fig. 3a), indicating that BC with clinico-pathological features of aggressiveness, such as TNBCs, display comparatively lower level of the tumor suppressor p140Cap. Similar results were obtained analyzing the distribution of p140Cap expression across the different molecular subtypes in the IEO cohort (see Methods) (Supplementary Fig. 3b). Overall, these findings further corroborate the notion of p140Cap as a tumor suppressor in BC and point to a possible involvement of p140Cap in influencing the TME immune infiltrate and the anti-tumor inflammatory response in BC patients.

### p140Cap rewires the immune tumor microenvironment to drive anti-tumor activity

Prompted by the above clinical observations, we investigated the involvement of p140Cap in influencing the immune infiltrate and inflammatory responses in the TME, by exploiting two syngeneic mouse models for human BC: (i) the TuBo cells, an established cell line from BALB/c-MMTV-NeuT mice, a model for HER2⁺ BC[35]; (ii) the highly tumorigenic and invasive murine mammary carcinoma cell line 4T1, a model for stage IV TNBC[36]. Both cell lines expresses basally low or even undetectable p140Cap levels and have previously revealed as amenable models to show the inhibitory effects on tumor progression and metastasis following restoration of p140Cap expression, in syngeneic mice[23]. We therefore expressed p140Cap in TuBo and 4T1 cells (Supplementary Fig. 4a, b) by retroviral transduction, and examined the qualitative and quantitative differences in the composition of the TME immune infiltrate in orthotopic mammary fat pad tumor outgrowths generated by p140Cap-expressing vs. mock TuBo and 4T1 cells in syngeneic BALB/c mice. p140Cap-expression in either cell line resulted in a significantly delayed tumor formation and an even more dramatic

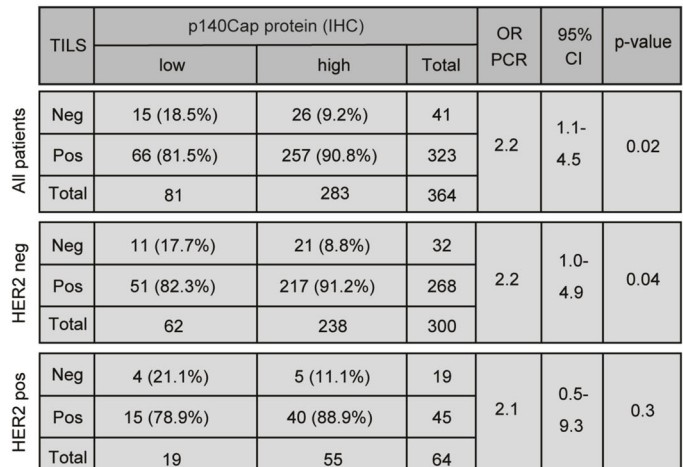

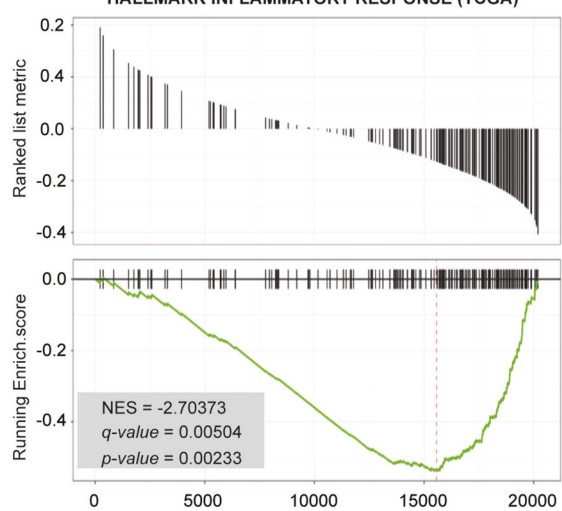

**Fig. 1 | A positive p140Cap status directly correlates with the presence of TILs and inversely with inflammatory hallmarks in human BC. a** Representative images of p140Cap and H&E staining, showing TILs, in HER2 negative and HER2 positive tumor breast tissues from 390 female patients. Black arrows point to TILs; red arrows indicate to cancer. Mag, magnifications. Bars, 200, 100 μm. **b** Quantitative analysis of the presence of TILs (positive or negative) according to p140Cap status (p140Cap$^{LOW}$, IHC score <1; p140Cap$^{HIGH}$, IHC score ≥1), evaluated on H&E stained TMA cores in All patients, as well as in HER2-negative and HER2-positive patients. OR odds ratio, CI confidence interval; *p*-value by Pearson's Chi-Squared Test. **c** GSEA[32,33] plot showing the behavior of the "Hallmark_Inflammatory_Response" Gene Set from MSigDB-5.0 in the cohort of 1095 BC female patients from TCGA [34], stratified according to *SRCIN1* transcript levels. The plot shows the distribution of genes in the set that are correlated or not with *SRCIN1* expression; two-tailed unpaired *t* test.

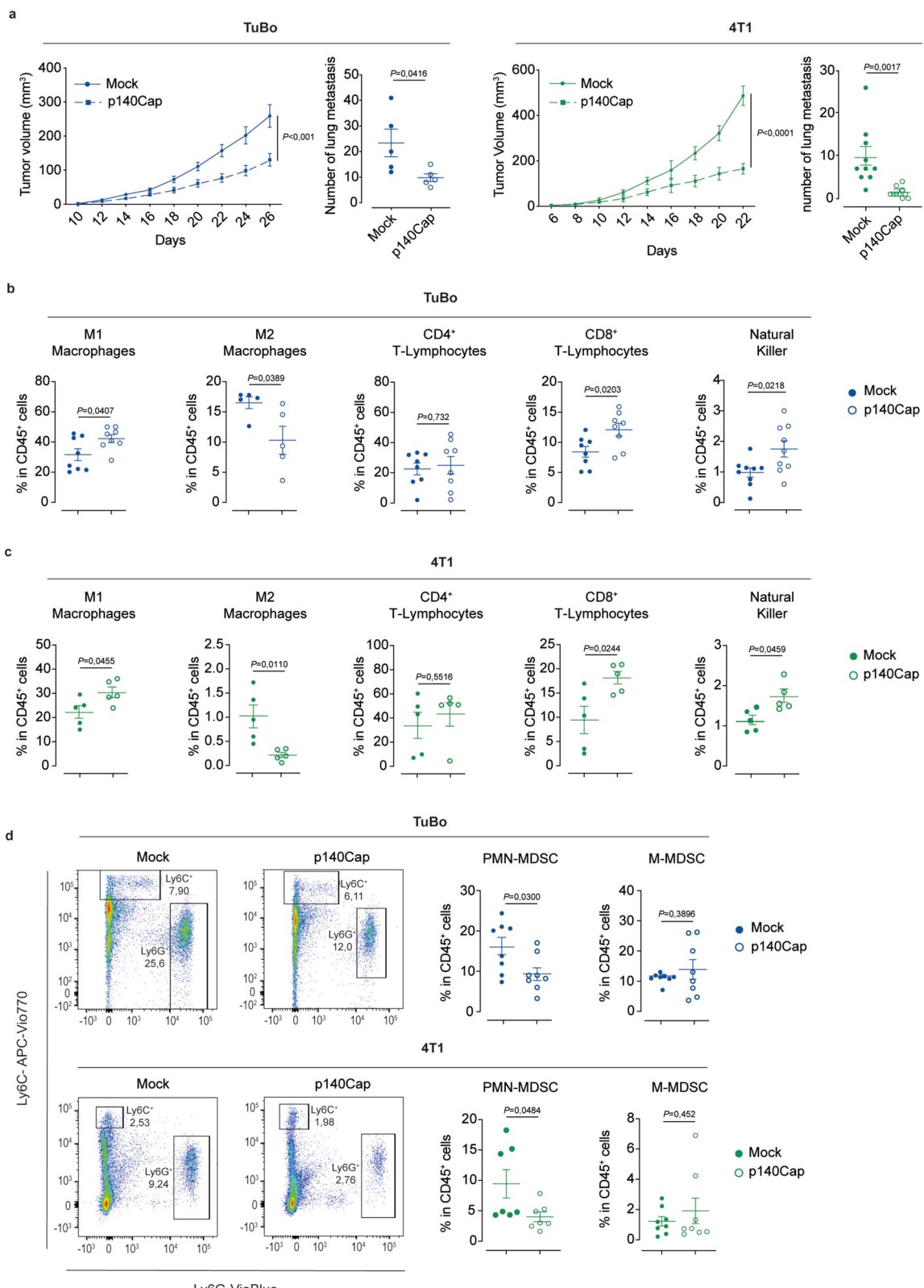

inhibition of metastatic spreading (Fig. 2a; Supplementary Fig. 4c, d). Fluorescence-activated cell sorting (FACS) analysis of dissociated tumor explants revealed that, compared to their mock controls, the immune infiltrate of p140Cap-expressing TuBo and 4T1 tumor outgrowths is characterized by: (i) a significantly higher TME infiltration by immune cells with a proinflammatory and anti-tumor activity, such as CD8+ T-lymphocytes and Natural Killer (NK) cells, with no significant differences in the sub-population of CD4+ T-lymphocytes (Fig. 2b, c); (ii) a statistically significant enrichment in anti-tumor M1-macrophages, accompanied by reduced infiltration of pro-tumorigenic M2-macrophages.

Collectively, these findings point to the existence of a causal link between p140Cap and TME infiltration by immune cells endowed with a cytotoxic anti-tumor activity, at the expense of tumor-promoting

**Fig. 2 | Reduced tumor growth and lung metastasis in p140Cap expressing cells and composition of the immune tumor microenvironment in mock and p140Cap tumors. a** Effect of p140Cap over-expression on tumor growth and metastasis in TuBo and 4T1 BC cell models. TuBo ($10^5$) or 4T1 ($10^4$) mock and p140Cap cells were inoculated into the fat pad of 8-weeks-old female BALB/c mice. Tumor growth was monitored and tumor size was measured every two days from tumor onset (TuBo, $n = 11$; 4T1, $n = 5$; data are represented for $n = $ x mices, two-tailed unpaired $t$ test). For metastasis analysis in TuBo mock and p140Cap tumor-bearing mice, tumors were surgically removed when they reached 10 mm diameter and mice were kept alive. After 5 weeks, mice were sacrified and lungs were explanted and analyzed. For metastasis analysis of 4T1 mock and p140Cap tumors-bearing mice, lungs were analyzed 22 and 30 days post-injection for mock and p140Cap cells, respectively. Representative dot plots show the number of lung metastasis as mean ± SEM (Standard Error of the Mean; TuBo, $n = 5$; 4T1, $n = 10$; 2way ANOVA). **b, c** Flow cytometry analysis for M1- and M2-macrophages, CD4$^+$ and CD8$^+$ T-Lymphocytes, Natural Killer cells in tumor-bearing mice. Representative dot plots show the percentage (%) of tumor infiltrated M1- and M2-macrophages, CD4$^+$ and CD8$^+$ T-Lymphocytes, Natural Killer (NK) cells, normalized on CD45$^+$ cells in TuBo mock and p140Cap tumor-bearing mice in panel (**b**) ($n = 8$/M1, CD4$^+$ and CD8$^+$ and $n = 9$/NK, $n = 5$/M2) and 4T1 mock and p140Cap tumor-bearing mice in panel (**c**) ($n = 5$/group). Data are represented for $n = $ x mice as mean ± SEM; two-tailed unpaired $t$ test. For TuBo mock and p140Cap tumors, the analysis was performed at day 26 or 32, respectively, while for 4T1 mock and p140Cap tumors at day 12. **d** Flow cytometry analysis for PMN-MDSCs (CD11b$^+$Ly6G$^+$Ly6C$^{low}$) and for M-MDSCs (CD11b$^+$Ly6G$^-$Ly6C$^+$) normalized on CD45$^+$ cells, in tumor-bearing mice. Representative dot plots show the percentage of tumor infiltrated PMN-MDSCs and M-MDSCs cells in TuBo and 4T1 mock and p140Cap tumor-bearing mice, as described in panels (**b**, **c**) (TuBo $n = 8$/group; 4T1 $n = 7$/PMN-MDSCs and $n = 8$/M-MDSCs). Data are represented for $n = $ x mice as mean ± SEM; two-tailed unpaired $t$ test.

immune cells. They are also consistent with data obtained from BC patients showing correlation between high p140Cap levels and increased TILs infiltrate. Notably, the heterogeneous population of myeloid-derived suppressor cells (MDSCs) plays a fundamental role in the establishment of an immunosuppressive TME by inhibiting anti-tumor activities of T-lymphocytes and NK cells, while promoting the recruitment/polarization of pro-tumor regulatory T-lymphocytes (Treg) and M2-like macrophages[37–39]. Therefore, dissociated tumor outgrowths were immunophenotypically characterized by FACS analysis using the CD11b$^+$/Ly6G$^+$/Ly6C$^{low}$ vs. CD11b$^+$/Ly6G$^-$/Ly6C$^+$ configuration to assess the relative distribution of polymorphonuclear (PMN)- vs. monocytic (M)-MDSCs precursors within the entire CD45$^+$ population (Fig. 2d). In both TuBo and 4T1 p140Cap tumors, a significant decrease in the percentage of PMN-MDSCs, compared to mock tumors was consistently observed, while no significant differences could be detected in the distribution of M-MDSCs (Fig. 2d). Considering that the CD11b$^+$Ly6G$^+$Ly6C$^{low}$ configuration does not allow to distinguish the immunosuppressive PMN-MDSCs precursors from pro-inflammatory neutrophils[37], and the lack of of specific neutrophil markers[40], we turned to an ex vivo approach to establish whether the subpopulation of CD11b$^+$Ly6G$^+$Ly6C$^{low}$ cells isolated from TuBo p140Cap-expressing tumors displayed immunosuppressive activity towards splenic CD8$^+$ T-lymphocytes. This assay clearly showed that the CD11b$^+$Ly6G$^+$Ly6C$^{low}$ cells purified from p140Cap-expressing tumors efficiently inhibit splenic CD8$^+$ T-lymphocyte proliferation, a phenotype expected of immunosuppressive PMN-MDSCs cells. In contrast, no substantial differences were observed in mock vs. p140Cap tumor-bearing mice, when we analyzed the suppressive activity associated with the bulk fraction of tumor-infiltrating CD11b$^+$ cells, or the bulk Ly6G$^+$ population isolated from the spleen (Supplementary Fig. 5a–d). These findings suggest that p140Cap exerts a selective loco-regional effect in counteracting accumulation/recruitment of immunosuppressive PMN-MDSCs in the stromal microenvironment and in secondary immune organs.

To directly evidence the implication of PMN-MDSCs in favoring tumor growth in the absence of p140Cap, we treated, twice a week by intraperitoneal injection, mice bearing mock-TuBo tumors with a monoclonal antibody (Mab), 1A8, against the Ly6G marker of PMN-MDSCs or with a vehicle control. Mice treated with 1A8-Mab showed a significantly impaired tumor growth compared to control mice. Of note, tumor growth inhibition achieved by selective targeting of PMN-MDSCs with 1A8-Mab was of the same magnitude of that obtained with p140Cap expression in TuBo cells (Supplementary Fig. 5e). FACS analysis confirmed that the 1A8-Mab specifically depletes the Ly6G$^+$ sub-population, in the primary tumor, and also in the blood and spleen of treated mice, which displayed a percentage of PMN-MDSCs similar to that found in TuBo-p140Cap tumor-bearing mice (Supplementary Fig. 5f). These data demonstrate that the CD11b$^+$Ly6G$^+$Ly6C$^{low}$ PMN-MDCs population has a tumor-promoting role in the BC TuBo model

and that its depletion is sufficient to curb tumor growth, with an effect similar to that observed with p140Cap expression. Overall, these results further support that p140Cap might exert a tumor growth inhibitory effect, shifting the overall profile of the TME from a tumor-promoting to a tumor-suppressing one by limiting the accumulation of immunoregulatory effectors such as PMN-MDSCs.

## p140Cap affects the secretion of the granulocyte-colony stimulating factor (G-CSF)

To highlight the possible mechanisms through which p140Cap may affect the composition of the TME immune infiltrate, we compared the transcriptomic profiles of mock- vs. p140Cap-expressing TuBo cells by RNA-Sequencing (RNA-Seq). Among the most significant differentially expressed genes, we found genes encoding for cytokines that variably influence the TME immune response by exerting either a pro-tumorigenic (*Csf3*)[41] or an anti-tumorigenic activity (*Il15*)[42,43], or ambivalent effects (*Tgfb1, Tgfb2 and Tgfb3*)[44,45]. In particular, p140Cap expression correlated with a remarkable decrease in mRNA for the pro-tumoral genes, *Csf3* and *Tgfb1*, concomitant with an increase of the anti-tumor gene, *Il15*. *Csf3*, encoding for the cytokine, Granulocyte-Colony Stimulating Factor (G-CSF), emerged as the top transcriptionally downregulated gene in TuBo-p140Cap cells vs. mock cells (Log$_2$FC: −1.075, $p$-value: $2.272 \times 10^{-46}$) (Fig. 3a). This cytokine is a major immune-related secreted factor that plays an important role in cancer progression[46,47], and its expression increases in different types of cancers, including BC[41], where G-CSF associates with poor prognosis[41,47,48]. The levels of the *Csf3* transcript and G-CSF protein in p140Cap-expressing vs. mock cells were validated by quantitative Real Time-PCR (RT-qPCR) and ELISA assay. In keeping with RNA-seq data, p140Cap expression invariably resulted in a significant reduction of the *Csf3* mRNA levels of 50% and 60% in TuBo and 4T1, respectively, compared to mock cells (Fig. 3b). Consistently, p140Cap expression caused a consistent reduction of G-CSF levels measured in the cell culture supernatant (Fig. 3b). Moreover, orthotopic mammary tumor xenografts of either TuBo-p140Cap or 4T1-140Cap cells revealed lower *Csf3* mRNA levels compared to tumors of equivalent size generated by their respective mock counterparts (Fig. 3c). Overall, these data show that p140Cap expression correlates with lower amounts of *Csf3* transcripts and ensuing G-CSF secretion, indicating that 140Cap might influence the stromal immune infiltrate composition due to its ability to interfere with the release of G-CSF.

## p140Cap expression affects the systemic distribution of PMN-MDSCs

G-CSF plays a fundamental role in the dynamic regulation of PMN-MDSCs in inflammation[49,50]. In cancer, G-CSF promotes accumulation of PMN-MDSCs in primary tumor lesions favoring not only their recruitment from the circulating pool, but also mobilizing them from the bone marrow[51]. In particular, the G-CSF released from the primary

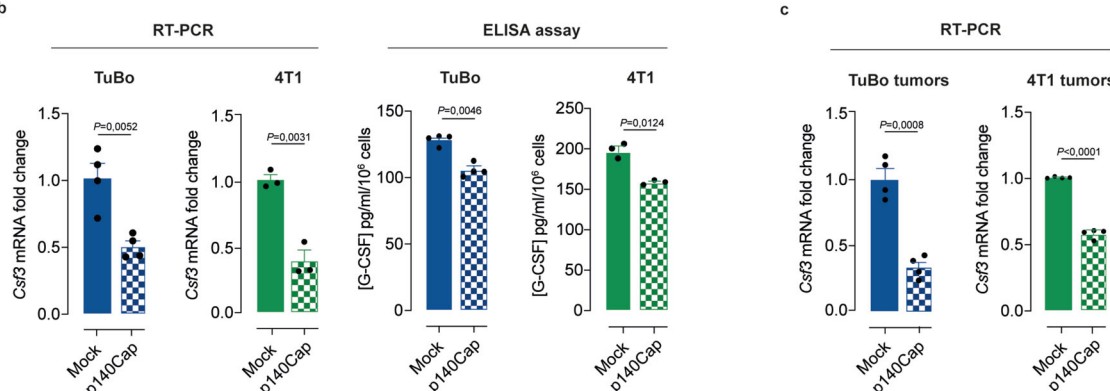

**a**

**HALLMARK_INFLAMMATORY_RESPONSE**

| Gene Name | Human Ortholog | log2FoldChange | lfcSE | pvalue | padj |
|---|---|---|---|---|---|
| Csf3 | CSF3 | -1.075 | 0.07516 | 2.272e-46 | 1.234e-44 |
| Tgfb3 | TGFB3 | 0.4484 | 0.05126 | 2.176e-18 | 2.585e-17 |
| Tgfb1 | TGFB1 | -1.547 | 0.3424 | 6.247e-06 | 2.292e-05 |
| Il15 | IL15 | 0.702 | 0.1915 | 0.0002474 | 0.0007256 |
| Tgfb2 | TGFB2 | 0.1917 | 0.07529 | 0.01089 | 0.02305 |

**Fig. 3 | G-CSF expression is strongly reduced in p140Cap positive BC model.** **a** RNASeq Data Analysis was performed on the transcriptomic profiles of mock- vs. p140Cap-expressing TuBo cells. Shown is the list of 18 differentially expressed genes (provided by BIOCARTA) involved in the inflammatory response upon p140Cap expression. **b**, **c** p140Cap modulates G-CSF expression and secretion. **b** G-CSF transcript was analyzed by quantitative real-time RT-PCR in mock and p140Cap TuBo ($n = 4$) and 4T1 cells ($n = 3$). G-CSF protein levels were measured in mock and p140Cap TuBo ($n = 4$) and 4T1 ($n = 3$) cell culture supernatants by ELISA. Data are represented for $n = x$ experimental repeats and shown as mean ± SEM; two-tailed unpaired $t$ test. **c** G-CSF transcript levels were analyzed by RT-PCR in mock and p140Cap TuBo and 4T1 tumors. Data are represented for $n = 4$ mice/group as mean ± SEM; two-tailed unpaired $t$ test.

tumor can trigger PMN-MDSCs mobilization directly from the bone marrow to the blood circulation, which, in turn, results in enhanced PMN-MDSCs recruitment to the primary site[52,53]. FACS analysis of blood and spleen samples from p140Cap TuBo or 4T1 tumor-bearing mice, compared to control animals, revealed a significantly reduced percentage of PMN-MDSCs in the bulk CD45[+] cell population isolated from these tissues. In contrast, comparable PMN-MDSCs levels were detected in the bone marrow of mock- vs. p140Cap-tumor bearing mice (Fig. 4a), likely due to the ability of the bone marrow to promptly replenish the pool of PMN-MDSCs by compensatory mechanisms linked to activation of the stem/progenitor compartment. Given the reported relevance of PMN-MDSCs function to the early phases of metastatization by favoring the formation of a metastatic niche[54], we analyzed the distribution of PMN-MDSCs in the lungs of 4T1 tumor-bearing mice. Preliminary time-course analysis of the kinetics of PMN-MDSCs lung infiltration in mock-4T1 tumor-bearing mice revealed that PMN-MDSCs could be recovered from dissociated lung tissues starting as early as 14 days post-injection, supporting their involvement in the formation of an early pre-metastatic lesion (Supplementary Fig. 6). Remarkably, the lungs of p140Cap-4T1 tumor-bearing mice showed a significant decrease in the presence of PMN-MDSCs, compared to the lungs of mock-4T1 tumor-bearing mice (Fig. 4b). ELISA analysis of sera of p140Cap- or mock-4T1 tumor-bearing mice was carried out to assess whether differences in the distribution of PMN-MDSCs at various organ sites could reflect alterations in G-CSF levels in the circulation. This analysis revealed that the G-CSF concentration was significantly lower in the serum of the p140Cap-tumor-bearing mice compared to control mice (TuBo: 11273 pg/ml vs. 7411 pg/ml; 4T1: 12981 pg/ml vs. 9305 pg/ml, in mock- vs p140Cap-tumors, respectively) (Fig. 4c).

Overall, these results point to a direct role of p140Cap in counteracting the accumulation of PMN-MDSCs into the blood and the spleen, associated with a reduction of circulating G-CSF, thus preventing PMN-MDSCs infiltration into the primary tumor site and their localization in pre-metastatic sites in distant organs.

## The decreased amount of G-CSF in p140Cap tumors depends on the suppressive function of p140Cap on the TIC compartment

Recent evidence in syngeneic mammary tumor models points to a key role of secreted G-CSF in mediating a pro-tumorigenic bidirectional cross talk between TICs and PMN-MDSCs. In particular, TICs might favor accumulation of immunosuppressive PMN-MDSCs in the TME through their unique ability to release G-CSF[55]. We therefore assessed whether the p140Cap-mediated reduction in G-CSF levels and the related effects on local and systemic PMN-MDCs dynamics, could be ascribed to a primary direct effect of p140Cap on the TIC compartment. To do this, we leveraged the mammosphere-forming assay in suspension conditions, a well-established proxy to measure stem cell self-renewal and frequency in mammary epithelial cell populations[5,6]. Consistent with previous reports[55], simply growing TuBo cells as mammospheres in suspension led to a statistically significant increase in G-CSF secretion compared to the 2D culture condition, further supporting the close association between enrichment in TICs and enhanced G-CSF production (Supplementary Fig. 7a). To address whether p140Cap could influence G-CSF secretion through regulation of the TIC compartment, we initially analyzed the consequences of p140Cap expression on the mammosphere-forming potential of TuBo and 4T1 cells, including in these studies the human BC cell line, MDA-MB-231, as a model of poorly differentiated TNBC BC with basally low p140Cap levels. We confirmed that, likewise TuBo and 4T1, also MDA-MB-231 cells, upon retroviral ectopic expression of p140Cap, display a significantly reduced *Csf3* mRNA transcription and G-CSF secretion, compared to mock cells (Supplementary Fig. 7b, c). In the mammosphere assay, TuBo, 4T1 and MDA-MB-231 cells invariably showed decreased mammosphere-forming efficiency as a consequence of

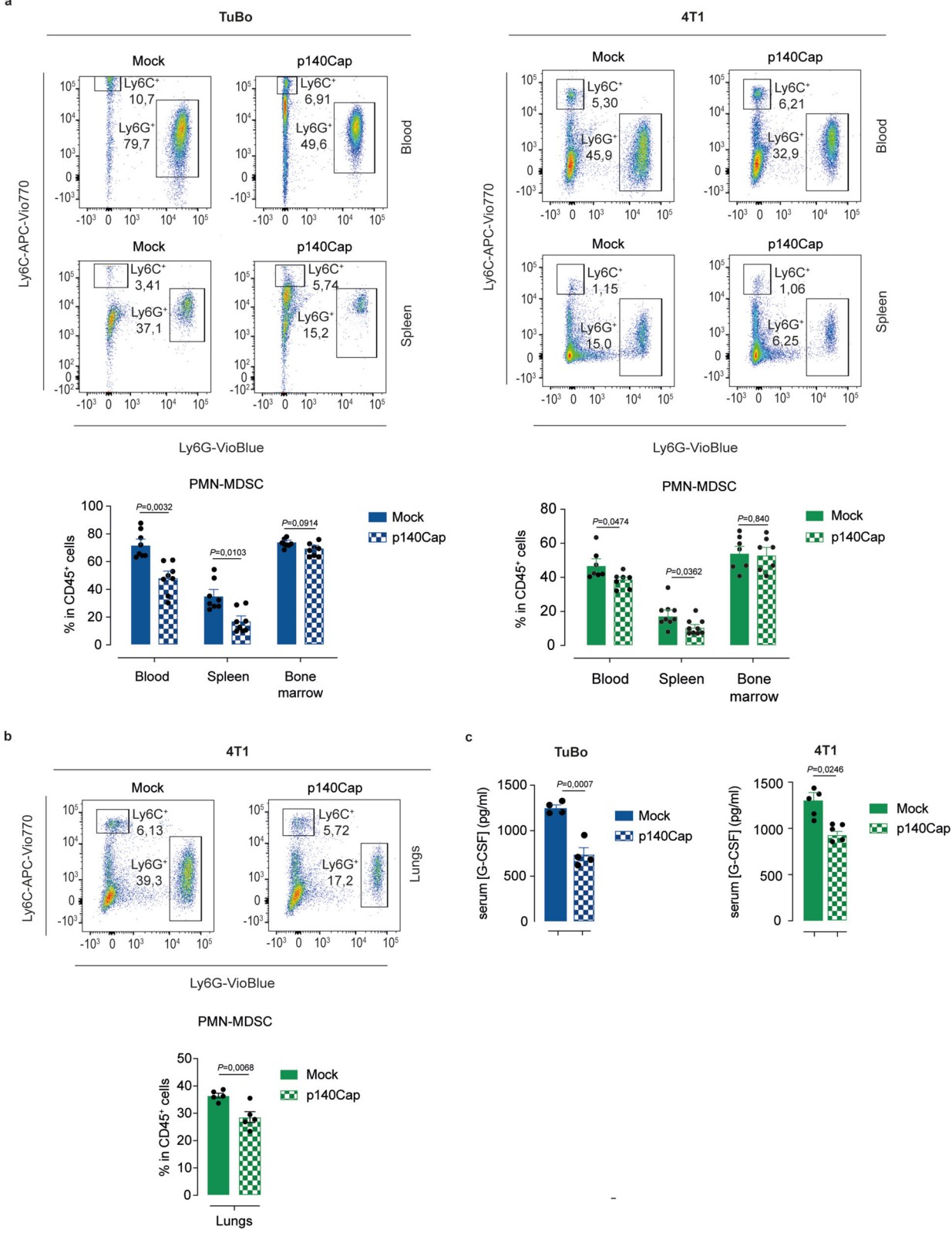

**Fig. 4 | p140Cap expression affects systemic distribution of PMN-MDSCs.**
**a**, **b** Flow cytometry analysis for PMN-MDSCs (CD11b⁺ Ly6G⁺ Ly6Cˡᵒʷ), normalized on CD45⁺ cells, in lungs, blood, spleen and bone marrow of tumor-bearing mice. Representative flow cytometry bar plots show: **a** the percentage of blood, spleen and bone marrow PMN-MDSCs in TuBo ($n = 8$) and 4T1 ($n = 7$/blood and bone marrow; $n = 9$/spleen) mock and p140Cap tumor-bearing mice; (**b**) the percentage of lung PMN-MDSCs in 4T1 mock and p140Cap tumor-bearing mice ($n = 5$ mice/group; data are represented for $n = x$ mice as mean ± SEM; two-tailed unpaired $t$ test). **c** G-CSF protein levels in sera of tumor-bearing mice. G-CSF protein levels were measured in sera collected from TuBo ($n = 4$) and 4T1 ($n = 5$) mock and p140Cap tumor-bearing mice by ELISA (Data are represented for $n = x$ mice as mean ± SEM; two-tailed unpaired $t$ test).

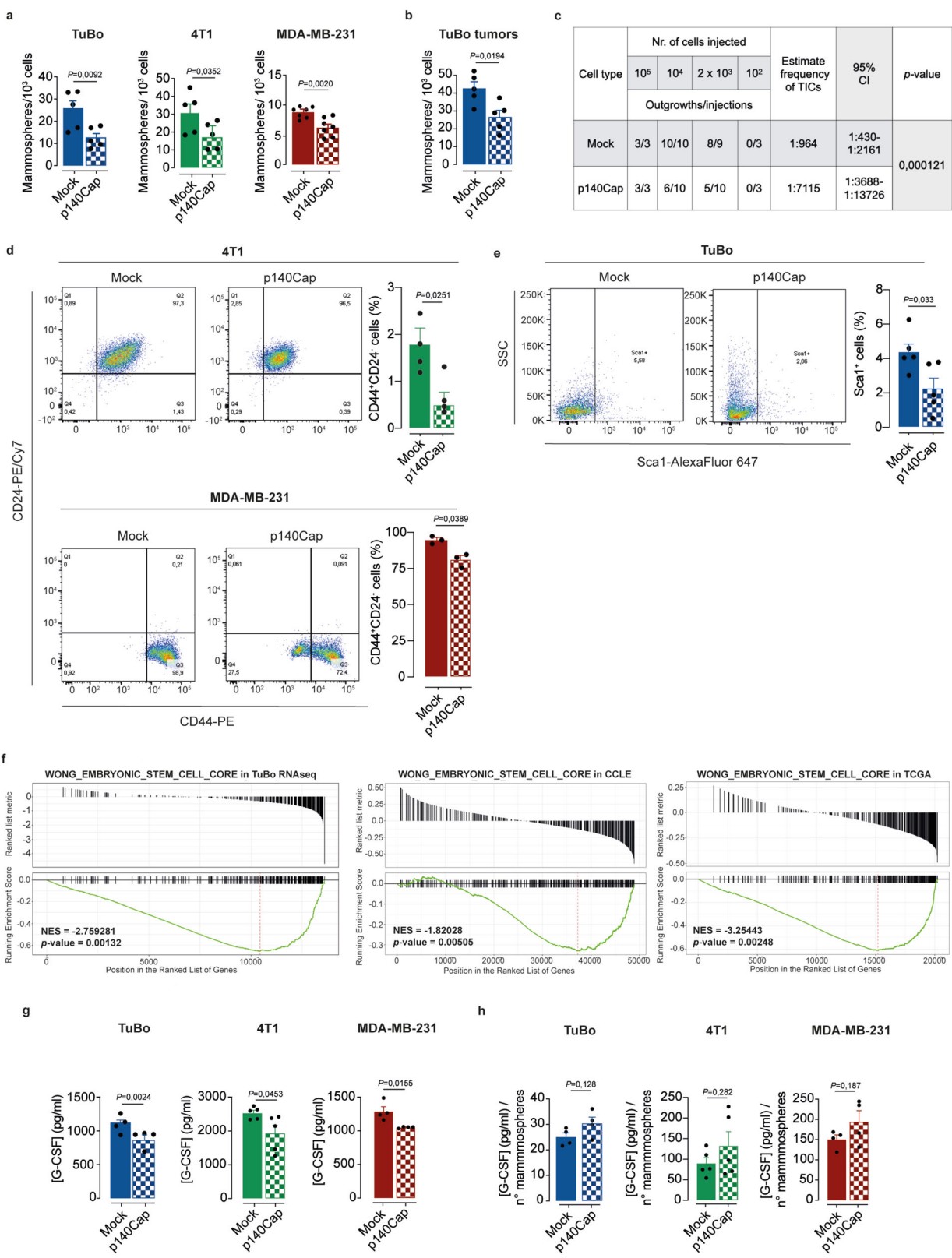

p140Cap expression, indicating a direct effect of p140Cap on TIC self-renewal ability (Fig. 5a; Supplementary Fig. 8). Similar effects were observed in cells freshly dissociated from p140Cap- vs. mock-TuBo tumors explants grown ex vivo as mammospheres (Fig. 5b). Moreover, we performed a limiting dilution xenograft assay (LDA), the golden standard to measure the frequency and tumorigenic potential of TICs in vivo. p140Cap- vs. mock-TuBo cells were injected at decreasing

concentration into the mammary fat-pad of syngeneic BALB/c mice and the formation of tumor outgrowths was monitored for 5 weeks post-transplantation. By Extreme Limiting Dilution Analysis (ELDA), we estimated a considerably lower frequency of TICs in p140Cap- vs. mock-TuBo cells (1 stem cell every 7115 injected cells vs. 1 stem cell every 964 cells, respectively, p-value 0,00012) (Fig. 5c). Collectively, these in vitro and in vivo functional studies point to a direct role of

**Fig. 5 | p140Cap impairs the breast TICs compartment. a, b** p140Cap effects on the Tumor-Initiating Cell (TIC) compartment. Mammosphere formation assay from panel (**a**) mock and p140Cap TuBo ($n = 5$), 4T1 ($n = 5$) and MDA-MB-231 ($n = 7$) cells and from panel (**b**) mock and p140Cap TuBo tumors ($n = 5$). Bar plots of mammosphere numbers are shown. Data are represented for $n = x$ experimental repeats and shown as mean ± SEM; two-tailed unpaired $t$ test). **c** In vivo Limiting Dilution Assay (LDA) of mock and p140Cap TuBo cells. Decreasing amounts (105, 104, 23 and 102) mock and p140Cap TuBo cells were injected into the mammary fat pad of 8-weeks-old female BALB/c mice. Tumor onset was monitored for 5 weeks post-transplantation. The Extreme Limiting Dilution Analysis (ELDA) software was used to calculate the LDA frequencies estimating the number of stem cells in the bulk population in mock and p140Cap TuBo tumors. **d, e** FACS analysis of stem cell/TIC markers. **d** FACS analysis of CD44+/CD24− cell populations in mock and p140Cap 4T1 ($n = 4$) and MDA-MB-231 ($n = 3$) cells. **e** FACS analysis of Sca1 expression in TuBo mock and p140Cap cells ($n = 5$). Bar plots of TIC marker quantification are shown. Data are represented for $n = x$ experimental repeats and shown as mean ± SEM; two-tailed unpaired $t$ test. **f** GSEA[32,33] plot of WONG_EMBRYONIC_STEM_CELL_CORE in TuBo RNAseq data, and in the CCLE[56] and TCGA[34] database. The plots show the distribution of genes correlated or not with SRCIN1 gene expression; two-tailed unpaired $t$ test. **g** G-CSF protein levels in mammosphere supernatants. G-CSF protein levels were measured in mock and p140Cap TuBo ($n = 4$), 4T1 ($n = 5$) and MDA-MB-231 ($n = 4$) mammosphere culture supernatants by ELISA. Data are represented for $n = x$ experimental repeats and shown as mean ± SEM; two-tailed unpaired $t$ test. **h** Bar plot represent the normalization of G-CSF (pg/ml), shown in panel (**g**), on mammosphere number generated from mock and p140Cap TuBo ($n = 4$), 4T1 ($n = 5$) and MDA-MB-231 ($n = 4$) cells shown in panel (**a**). Data are represented for $n = x$ experimental repeats, two-tailed unpaired $t$ test.

p140Cap in affecting the number and tumorigenic potential of TICs in BC. Accordingly, FACS analysis revealed, significant reduction in the pool of cells displaying stemness traits, as witnessed by the decrease in Sca1+ cells in the bulk p140Cap-TuBo cells population, and CD44+/CD24− cells in the bulk p140Cap- 4T1 and MDA-MB-231 cell populations (Fig. 5d, e). GSEA[32,33] meta-analysis of the transcriptomic profiles of 51 BC cell lines derived from the Cancer Cell Line Encyclopedia (CCLE)[56] and the TCGA[34] database, revealed that p140Cap is inversely correlated with the expression of sets of genes associated with self-renewal (WONG_EMBRYONIC_STEM_CELL_CORE, provided by the MSIGDB database) (Fig. 5f). In the supernatants of mammosphere cultures generated by p140Cap- vs. mock-TuBo, 4T1 and MDA-MB-231 cells, p140Cap expression was invariably associated with a reduction in mammosphere-forming efficiency, paralleled by a consistent decrease in the total amount of secreted G-CSF (Fig. 5g). This result pointed to a direct correlation between p140Cap-induced shrinkage of the TIC compartment, assessed in the mammosphere assay, and impaired G-CSF secretion, as confirmed by the ratio between total G-CSF levels and mammosphere number (Fig. 5h).

## p140Cap controls the TIC compartment through inhibition of the Wnt/β-Catenin pathway

The Wnt/β-Catenin pathway has emerged as a major responsible for the appearance and maintenance of TICs in several cancer types[57]. While the direct interaction between p140Cap and β-Catenin has been previously described[25,26], the functional significance of this interaction and its actual relevance to BC remains elusive. We reasoned that the suppressive control of p140Cap over the TIC compartment could be mediated by its ability to interact with and negatively regulate β-Catenin activity. Supporting this hypothesis, GSEA analysis[32,33] of RNAseq data from p140Cap- vs. mock-TuBo cells unraveled an inverse correlation between p140Cap expression and transcript levels of a set of β-Catenin target genes comprised in the FEVR_CTNNB1_TARGETS_DN signature (Fig. 6a). A similar inverse correlation was observed in the CCLE[56] and TCGA[34] databases (Fig. 6a). To provide formal proof that inhibition of β-Catenin is the upstream event in the negative regulation of TICs by p140Cap, we measured the activation status of β-Catenin in mammosphere cultures of p140Cap- vs. mock-TuBo and 4T1 cells, as well as in bulk p140Cap- vs. mock-TuBo tumor outgrowths. WB with a specific antibody directed against the active form of the β-Catenin (i.e. not inactivated by phosphorylation on residues Ser33, Ser37 and Thr41)[10] revealed a strong reduction of active unphosphorylated β-Catenin in mammospheres derived from both p140Cap-TuBo (Fig. 6b, c) and p140Cap-4T1 (Fig. 6d) cells and tumor outgrowths, compared to their respective mock counterparts. These results point to a primary role of p140Cap in the process of β-Catenin phosphorylation, a prerequisite for its ubiquitination and ensuing proteasomal degradation, as the mechanism underlying p140Cap-mediated inhibition of β-Catenin activity in the TIC compartment. Stably expression in p140Cap-TuBo cells of a constitutively active, non-

phosphorylatable form of β-Catenin, carrying the S33Y mutation[58] (Fig. 6e), restored the mammosphere-forming efficiency to the levels observed in mock-TuBo cells, with a concomitant restoration of *Csf3* transcription and G-CSF secretion (Fig. 6f–i), demonstrating that a constitutively active β-Catenin is sufficient to overcome the inhibitory effects of p140Cap on the TIC compartment and G-CSF secretion. In vivo, compared to control p140Cap-TuBo cells, p140Cap-TuBo cells expressing the active S33Y-mutant β-Catenin showed accelerated tumor growth and increased PMN-MDSCs infiltration in the primary tumor, associated with higher PMN-MDSCs concentration in the blood and spleen, thereby recapitulating the typical condition of mock TuBo-injected mice (Fig. 6j, k). The sum of these findings point to the existence of a complex epistatic axis where negative regulation of β-Catenin activity by p140Cap is the upstream event leading to restriction of the TIC compartment and consequent reduction of Csf3 transcription and G-CSF secretion. p140Cap inhibition of this axis ultimately prevents the creation of a tumor conducive microenvironment by counteracting the peripheral mobilization and intra-tumor infiltration of immunosuppressive PMN-MDSCs.

## p140Cap regulates the β-Catenin active pool through stabilization of the destruction complex

In search for the mechanism underlying the inhibitory control of p140Cap over the epistatic β-Catenin/TIC/G-CSF/PMN-MDSCs axis, we hypothesized that p140Cap inhibition of β-Catenin activity might take place in the context of the so-called "destruction complex", a multiprotein machinery that controls β-Catenin phosphorylation and its ensuing proteasomal degradation. This complex includes the tumor suppressors Axin1, the Adenomatous Polyposis Coli (APC), the Ser/Thr kinases GSK-3β and CK1, the protein phosphatase 2 A (PP2A), and the E3-ubiquitin ligase β-TrCP[59]. We therefore initially investigated whether p140Cap could physically interact with some components of the complex, performing immunoprecipitation experiments from dissociated mammospheres from both TuBo and 4T1 cells. In addition to the previously reported p140Cap/β-Catenin interaction[25], we found that p140Cap co-immunoprecipitates with Axin1 and GSK-3β (Fig. 7a, b). Even more interestingly, upon β-Catenin immunoprecipitation, higher amounts of both Axin1 and GSK-3β were recovered comparing p140Cap-TuBo vs. mock mammospheres (Fig. 7c). Likewise, in either GSK-3β or Axin1 immunoprecipitates, we detected higher levels of β-Catenin in p140Cap vs. mock mammospheres (Fig. 7d, e). Altogether, these immunoprecipitation studies, in addition to demonstrating that p140Cap can enter the multiprotein destruction complex, argue that its presence enhances the stoichiometric interaction of β-Catenin with the other components of the destruction machinery, likely leading to its stabilization and enhanced inactivation. Accordingly, in the presence of the proteasome inhibitor MG132, we observed higher levels of the ubiquitinated form of β-Catenin in mammospheres from p140Cap- vs. mock-TuBo cells (Fig. 7f), further supporting a direct role of

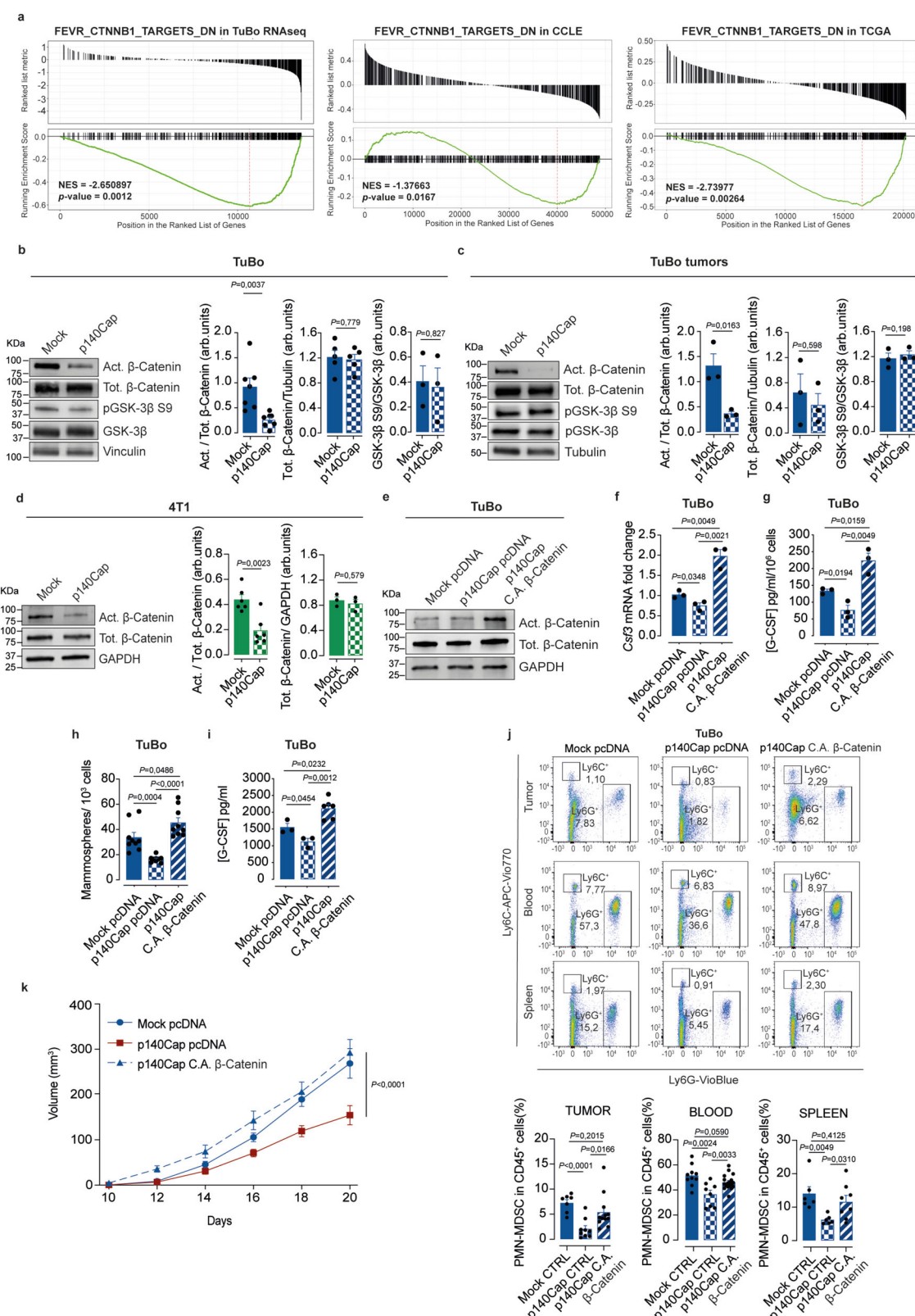

p140Cap in promoting β-Catenin inactivation acting in the context of the destruction complex. We next set out to formally demonstrate the interaction of p140Cap with the different component of the destruction complex directly in living cells. To this aim, we leveraged the BioID assay, a proximity labeling method that relies on fusing a bait protein to a promiscuous biotin ligase, the BirA enzyme, allowing the detection of protein-protein interactions in a range from 10

to 20 nm[60]. We used a fusion p140Cap-BirA-HA protein, where the BirA enzyme was located at the C-terminal p140Cap, as described in[61](Fig. 7g). Following expression of the fusion p140Cap-BirA-HA protein, and an empty BirA-HA protein as a negative control, together with Axin1-GFP, in HEK293 cells, we found that p140-BirA-HA efficiently biotinylated Axin1, total β-Catenin, and GSK-3β. Most importantly, this assay revealed that, in addition to the presence of

**Fig. 6 | p140Cap affects the TIC compartment through reduction of the β-Catenin active form. a** GSEA[32,33] plot of FEVR_CTNNB1_TARGETS_DN in TuBo RNAseq, CCLE[56] and TCGA[34] database. Plots show the distribution of genes correlated or not with SRCIN1 gene expression; two-tailed unpaired *t* test. **b–d** Effect of p140Cap expression on β-Catenin activation. Immunoblot analysis in mammospheres from (**b**) mock and p140Cap TuBo cells, (**c**) tumors. Vinculin,Tubulin and GAPDH were used as loading controls. Act = active β-catenin ($n = 7$/cells, $n = 3$/tumors experiments), Tot = Total β-catenin ($n = 5$/cells, $n = 3$/tumors), GSK-3β S9 ($n = 3$/cells and tumors). **d** mock and p140Cap 4T1 cells. Act = active β-catenin ($n = 6$), Tot = Total β-catenin ($n = 3$); arb.units = arbitrary units. Data are representative of $n = x$ experimental repeats and shown as mean ± SEM; two-tailed unpaired *t* test. In (**b**) total β-catenin and GSK−3β S9 have been run on a separate blot. In (**c**) active and total β-catenin have been run on a separate blot. In (**d**) total β-catenin has been run on a separate blot. **e** Generation of p140Cap TuBo cells expressing a constitutive active (C.A.) form of β-Catenin or empty pcDNA vector, as control. C.A. expression evaluated by immunoblotting. GAPDH: loading controls. Total β-catenin has been run on a separate blot. Data are represented for $n = 3$ experimental repeats. **f, g** C.A. β-Catenin effect on G-CSF expression in p140Cap cells. **f** G-CSF mRNA by RT-PCR ($n = 3$) and (**g**) G-CSF protein levels by ELISA ($n = 3$).

Data are represented for $n = x$ experimental repeats and shown as mean ± SEM; two-tailed unpaired *t* test. **h** Mammosphere formation assay from mock pcDNA, p140Cap pcDNA and p140Cap C.A β-Catenin in TuBo cells. Bar plots of mammosphere number are shown. Data are represented for $n = 9$ experimental repeats; data presented as mean ± SEM; two-tailed unpaired *t* test. **i** G-CSF protein levels were measured in mock pcDNA ($n = 3$), p140Cap pcDNA ($n = 3$) and p140Cap β-Catenin C.A ($n = 6$). TuBo mammosphere supernatants by ELISA. Data are represented for $n = x$ experimental repeats and shown as mean ± SEM; two-tailed unpaired *t* test. **j** Representative bar plots show the percentage of PMN-MDSC cells normalized on CD45 + cells, in tumor, blood and spleen of mock pcDNA, p140Cap pcDNA and p140Cap C.A β-Catenin. TuBo pcDNA tumor-bearing mice ($n = 7$/tumor, $n = 10$/blood, $n = 6$/spleen); TuBo p140Cap pcDNA tumor-bearing mice ($n = 9$/tumor, $n = 8$/blood, $n = 6$/spleen); TuBo p140Cap C.A β-Catenin tumor-bearing mice ($n = 12$/tumor, $n = 18$/blood, $n = 8$/spleen). Data are represented for $n = x$ mice as mean ± SEM; two-tailed unpaired *t* test. **k** C.A. β-Catenin expression on tumor growth. $10^5$ pcDNA ($n = 8$), p140Cap pcDNA ($n = 8$) and p140Cap C.A. β-Catenin ($n = 13$) cells were injected into 8-weeks-old female BALB/c mice. Tumor size was measured every two days. Data are represented for $n = x$ mice as mean ± SEM; 2way ANOVA.

biotinylated β-Catenin associated with p140-BirA-HA, compared to the control empty BirA-HA protein, the biotinylated β-Catenin appeared in the form of the triple phosphorylated inactive β-Catenin, the functional post-translational modification executed by the coordinated action of CK1 and GSK-3β in the context of the destruction complex to prime β-Catenin ubiquitination and proteosomal degradation (Fig. 7h). Therefore, in addition to provide formal evidence of the direct interaction of p140Cap with the different components of the destruction machinery directly in living cells, these findings argue that, in the presence of p140Cap, β-Catenin resides in the destruction complex predominantly in its inactive form.

It has been recently shown that the formation of the destruction complex is driven by protein liquid-liquid phase separation (LLPS) of Axin1[62], due to its intrinsic property to share features with known components of biomolecular condensates, including the ability to form membraneless spherical punctate structures in cells[63]. Phase separation of Axin1 in living cells has recently emerged as an efficient mechanism favoring the assembly of the β-Catenin destruction machinery in spatially confined compartments, the LLPS punctae, where the biochemical reactions controlling the activation state of β-Catenin pathway take place, in particular its phosphorylation[62,64,65] We therefore asked whether p140Cap could impact on the dynamics of Axin1-dependent LLPS punctae. Following transient transfection of Axin1-GFP, p140Cap-TuBo cells showed a significantly increased steady-state number of LLPS punctae/cell, compared to Axin1-GFP transfected mock-TuBo cells (Fig. 8a). Furthermore, in TuBo cells concomitantly transfected with Axin1-GFP and p140Cap-RFP, we detected a very high frequency of colocalization of the two overexpressed proteins in LLPS punctae (Fig. 8b). In FRAP experiments conducted in mock-TuBo cells transfected with Axin1-GFP, alone or in combination with p140Cap-RFP, we observed a significant delay in Axin1-GFP fluorescence recovery in cells expressing p140Cap-RFP vs. control cells (Fig. 8c). This was associated with a concomitant increase in the percentage of immobile Axin1-GFP. Altogether, these immunofluorescence imaging studies indicate that the presence of p140Cap can positively influence the kinetics of LLPS punctae formation, while concomitantly reducing the mobility of Axin1 in these structures. These results are in keeping with our biochemical analyses (see Fig. 7) indicating a crucial role of p140Cap in promoting β-Catenin inactivation through stabilization of the destruction complex and enhanced β-Catenin ubiquitination. Arguably, pharmacological interventions mimicking the function of p140Cap at the level of the destruction complex might represent an efficient strategy to selectively target an aberrant β-Catenin activity in cancer.

## Pharmacological stabilization of the destruction complex results in β-Catenin inhibition

To provide proof of evidence in support of this idea, we tested whether the pharmacological Axin1 stabilizer IWR-1 that increases the stability of the β-Catenin destruction machinery[66], could mimic p140Cap function in the destruction complex and its effect on tumor phenotypes. We found that treatment of mammospheres from naïve TuBo cells with IWR-1 resulted in higher amounts of both Axin1 and GSK-3β co-immunoprecipitating with β-Catenin (Fig. 9a), indicating that, likewise p140Cap expression, Axin1 stabilization by IWR-1 is sufficient per se to increase the physical interaction of β-Catenin with other components of the destruction machinery. Consistently, in naïve TuBo cells, IWR-1 treatment also fully recapitulated the inhibitory effects observed with p140Cap expression in terms of inhibition of mammosphere-forming ability and reduced secretion of G-CSF associated with a decrease in active β-Catenin levels (Fig. 9b–d). In vivo treatment of mock-TuBo tumor-bearing mice with IWR-1 led to a significant delay in tumor growth kinetics, similar to that observed in p140Cap-TuBo tumor-bearing mice (Fig. 9e). Likewise, mock-TuBo tumor-bearing mice treated with IWR-1, compared to vehicle-treated mice, showed reduced G-CSF serum levels, accompanied by decreased local PMN-MDSCs infiltration of the primary tumor TME, as well as reduced systemic PMN-MDSCs levels in the blood and spleen, thus fully recapitulating the conditions observed in p140Cap-TuBo tumor-bearing mice (Fig. 9f, g). The congruency of results observed with IWR-1 and p140Cap expression in TuBo cells in vitro and in vivo strongly argues that pharmacological interventions that stabilize the β-Catenin destruction complex, as exemplified by the use of IWR-1 in naïve TuBo cells, might represent an additional therapeutic strategy to selectively interfere with the β-Catenin/TIC/G-CSF/PMN-MDSCs axis, thereby restoring an efficient immune response to prevent tumor progression and metastasis. They also argue for the potential use of p140Cap as a biomarker to stratify those BC patients who, by showing very low or undetectable p140Cap expression in their primary tumor, might be considered eligible for therapeutic intervention with drugs able to stabilize the β-Catenin destruction complex.

## Discussion

This study highlights a function of the p140Cap protein in orchestrating an efficient anti-tumor response in the tumor microenvironment of breast tumors. This notion expands current knowledge of p140Cap as a tumor suppressor, a function found to be relevant to different types of cancer in previous studies, in particular neuroblastoma and BC[23,67]. The tumor suppressor function of p140Cap has been historically ascribed to its unique

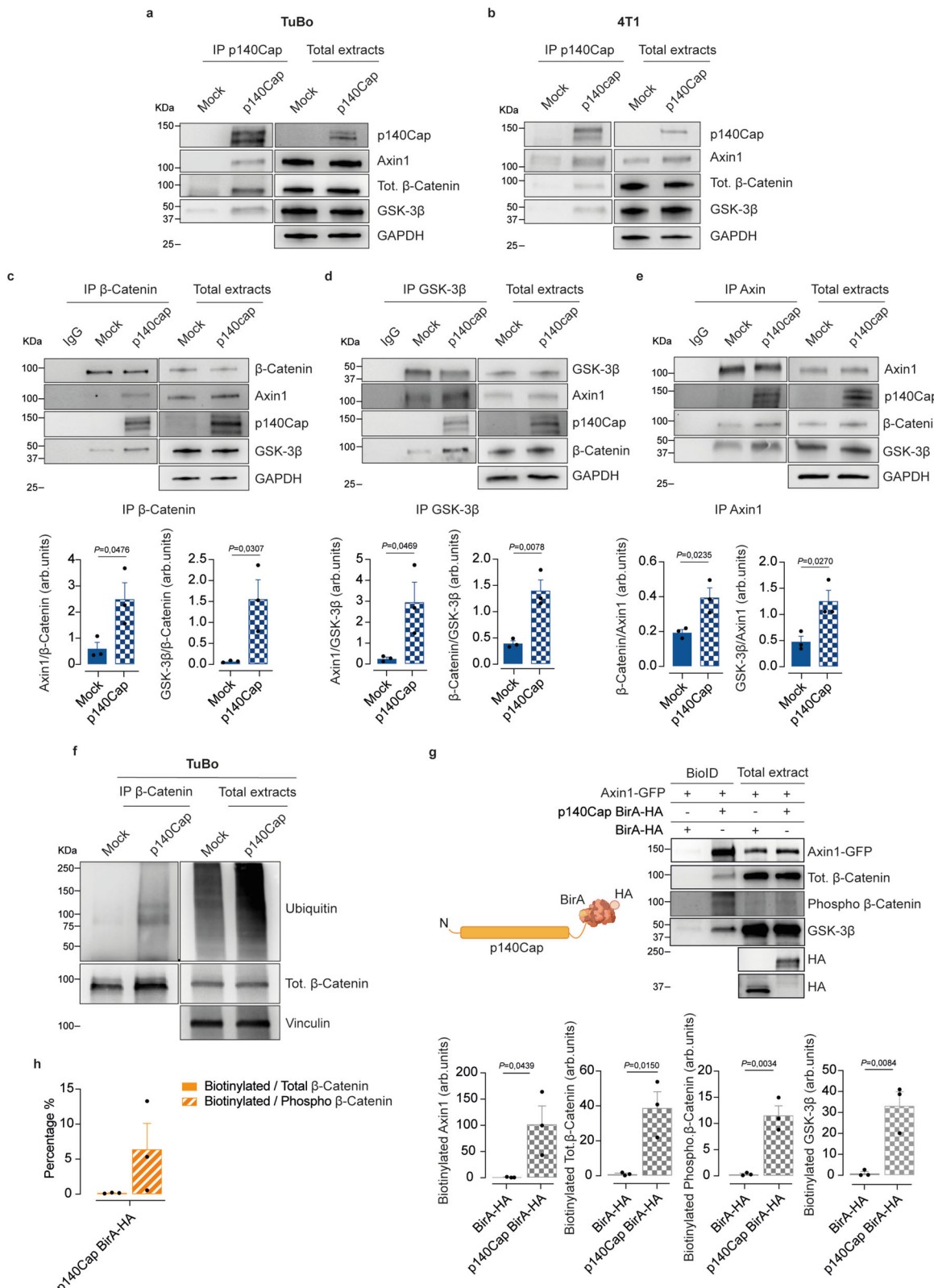

ability to behave as an adaptor protein that interferes with the activity of several oncogenic circuitries, including receptor and non-receptor tyrosine kinase pathways, for instance Src, Egfr and Erbb2 signaling, and small GTPase signaling, involving Rac1 and Tiam1, thereby down-modulating key cancer-relevant pheno-types, foremost proliferation and invasion, involved in tumor progression and metastasis[24]. It is not surprising therefore that, as

observed in BC patients, low p140Cap expression predicts a more severe prognosis[23,24]. Here, by the integration of functional and molecular data obtained leveraging preclinical murine and human breast tumor models, we reveal that, in addition to its cell-autonomous ability to negative modulate oncogenic signaling pathways, p140Cap also influences cell-extrinsic events by sup-pressing a tumor permissive immune response in favor of an

**Fig. 7 | p140Cap is a component of the destruction complex. a–e** Effect of p140Cap expression on the Destruction Complex stability.
**a**, **b** Immunoprecipitation of p140Cap and immunoblot analysis with antibodies to β-Catenin, Axin1, GSK−3β in mock vs. p140Cap TuBo in (**a**) and mock vs. p140Cap 4T1 mammospheres in (**b**) ($n = 3$). Data are representative of $n = x$ experimental repeats. **c**−**e** Immunoprecipitation of β-Catenin in (**c**), GSK−3β in (**d**), Axin1 in (**e**), or IgG (negative control) from mock and p140Cap TuBo mammospheres, and immunoblot analysis with antibodies to p140Cap, β-Catenin, GSK−3β, and Axin1. Bar plots are represented for $n = 3$ experimental repeats and shown as mean ± SEM; two-tailed unpaired $t$ test). GAPDH was used as loading control; arb.units = Arbitrary Units. **f** Immunoprecipitation of β-Catenin from Mock and p140Cap TuBo mammospheres treated with MG132 followed by immunoblot analysis with

antibodies to ubiquitin and β-Catenin. Vinculin was used as loading control. Data are represented for $n = 3$ experimental repeats. **g** BioID assay in HEK293 cells. Extracts from HEK293 cells transfected with p140Cap-BirA-HA and BirA-HA constructs in combination with Axin1-GFP were isolated with the BioID protocol and blotted with antibodies to Axin1, Total β-Catenin, Phospho β-Catenin, GSK-3β and HA. BirA-HA construct was used as negative control. Data are represented for $n = 3$ experimental repeats; bar plot represent biotinylated protein normalized on HA transfected amount as mean ± SEM; two-tailed unpaired $t$ test. **h** Bar plot represents the percentage of biotinylated total or Phospho β-Catenin normalized on the total amount of total or Phospho β-Catenin, respectively. Data are represented for $n = 3$ experimental repeats and shown as mean ± SEM; two-tailed unpaired $t$ test.

efficient anti-tumor response in the TME of the primary tumor. Indeed, we show that p140Cap expression associates with a TME characterized by enhanced homing of proinflammatory and anti-tumor immune cells, such as CD8$^+$ T-lymphocytes, NK cells and M1-macrophages, at the expense of cells endowed with immunosuppressive, tumor-promoting activities, such as PMN-MDSCs, regulatory T cells (Tregs) and M2-macrophages. These results are of the utmost relevance to a deeper understanding of the function of p140Cap as a tumor suppressor, considering the emerging liaison between TME anti-tumor immune response and negative control of tumor progression and metastasis[68] and, conversely, the close association between an immunosuppressive TME, with presence of MDSCs, Tregs and M2-polarized macrophages, and enhanced tumor progression[69–71].

Mechanistically, we show that p140Cap operates upstream of an epistatic functional axis in which its primary action is the downregulation of the TIC compartment present in the bulk tumor population. This results in reduced production of the inflammatory cytokine G-CSF, which is notably implicated in the stromal recruitment and infiltration by immunosuppressive PMN-MDSCs[49,50]. More importantly, in syngeneic mouse models in vivo, we show that p140Cap inhibition of the TIC compartment, in addition to resulting in impaired tumor growth, G-CSF secretion and stromal PMN-MDSCs infiltration at the primary tumor site, also has long distance inhibitory effects on the metastatic process. This is due to the reduced systemic levels of G-CSF that cause impaired mobilization of PMN-MDSCs from the bone marrow to the blood circulation. As a consequence, reduced amount of circulating PMN-MDSCs are available for stromal recruitment in the primary tumor site, where they promote tumor immune escape, as well as for the infiltration of distant organ sites, where they have been reported to favor the formation of a pre-metastatic niche enhancing tumor cell colonization[54,72,73]. Arguably, these effects are also due to the reduced availability of local and systemic G-CSF required for PMN-MDSCs chemotaxis[52,53]. Not surprisingly, therefore, PMN-MDSCs have been found to be associated with poor overall and progression-free survival in the meta-analysis of various types of solid tumors[69,74], and their presence in the blood and in the stromal tumor infiltrate has been considered as a potential marker for prognostic prediction and therapeutic target for cancer therapy[75,76].

Mechanistically, we reveal that the downregulation of the entire TIC/G-CSF/PMN-MDSCs axis depends on the ability of p140Cap to inhibit the activity of β-Catenin, which is notably a major determinant of TIC self-renewal and tumorigenic potential[57]. A role for the β-catenin signaling pathway in fine tuning the granulocytic production has also been described[77]. Indeed, here, by integrating biochemical studies and imaging analyses in living cells, we provide converging evidence pointing to a direct involvement of p140Cap in stabilizing the multiprotein β-Catenin destruction complex, thereby promoting the key inactivating post-translational modifications required to extinguish the β-Catenin signaling pathway. In particular, we reveal that: (i) p140Cap physically interacts with several components of the β-Catenin destruction machinery, including β-Catenin itself, Axin1 and GSK3β; (ii)

p140Cap is in close proximity with the inactive form of β-Catenin, phosphorylated by the sequential action of CK1 and GSK3β on residues Ser33, Ser37 and Thr41, which is a prerequisite for its subsequent ubiquitination and proteasomal degradation[59]; (iii) p140Cap co-localizes with Axin1 in discrete subcellular compartments, the LLPS punctae, which have been reported to constitute signaling platforms where the biochemical reactions controlling the β-Catenin pathway activation take place[62,65]; (iv) p140Cap expression enhances the kinetics of LLPS punctae formation and promotes Axin1 immobilization in their context. In keeping with these findings, p140Cap expression in the TuBo murine mammary tumor model leads to enhanced β-Catenin ubiquitination. Collectively, the sum of our biochemical and imaging studies highlights a tumor suppressor function of p140Cap as a direct negative regulator of β-Catenin, revealing that the stabilization of the β-Catenin destruction machinery represents the bona fide mechanistic site of action for the p140Cap-mediated inhibition of the epistatic β-Catenin/TIC/G-CSF/PMN-MDSCs axis.

The sum of the findings of this study are of the utmost importance from a clinical standpoint. Arguably, pharmacological interventions directed to the stabilization of the β-Catenin destruction complex might represent an amenable therapeutic strategy to counteract the formation of a tumor conducive immune response involved in local tumor progression and distant metastatic diffusion in those breast tumors, exemplified by BC tumors with low p140Cap expression, that display alteration of the β-Catenin signaling pathway. We provide proof of concept of this, using a drug, IWR-1, which acts as a pharmacological Axin1 stabilizer that increases the stability of the β-Catenin destruction machinery[66]. Indeed, this drug mimics the tumor suppressor function of p140Cap on the epistatic β-Catenin/TIC/G-CSF/PMN-MDSCs axis, recapitulating in its entirety both the effects of p140Cap at the primary tumor site (shrinkage of the TIC compartment and inhibition of tumorigenicity) and its tumor-extrinsic effects at the local and systemic level (reduced secretion of G-CSF and impaired PMN-MDSCs systemic mobilization and local TME infiltration).

Further supporting the clinical potential of these findings, we report in this study the results from a large consecutive cohort of real-life BC female patients that, in addition to the previously described association between proficient p140Cap expression in the primary tumor site and a more favorable prognosis in BC[23,24], show that: (i) low p140Cap levels associate with a remarkably higher frequency to the triple-negative molecular subtype, typically characterized by intrinsic biological aggressiveness and poor prognosis; (ii) p140Cap expression in the primary tumor correlates with enrichment in stromal TILs; (iii) the presence of TILs correlates with improved overall survival, a finding consistent with accumulated evidence supporting the clinical value of stromal TIL determination as a favorable prognostic biomarker in both neoadjuvant and adjuvant BC patients[30]. Consistently in silico analysis of the TGCA cohort of BC patients shows correlation between genes associated with inflammatory and *SRCIN1*/p140Cap transcriptional levels.

In summary, the bulk of preclinical and clinical findings of this study expand our current understanding of the function of p140Cap in

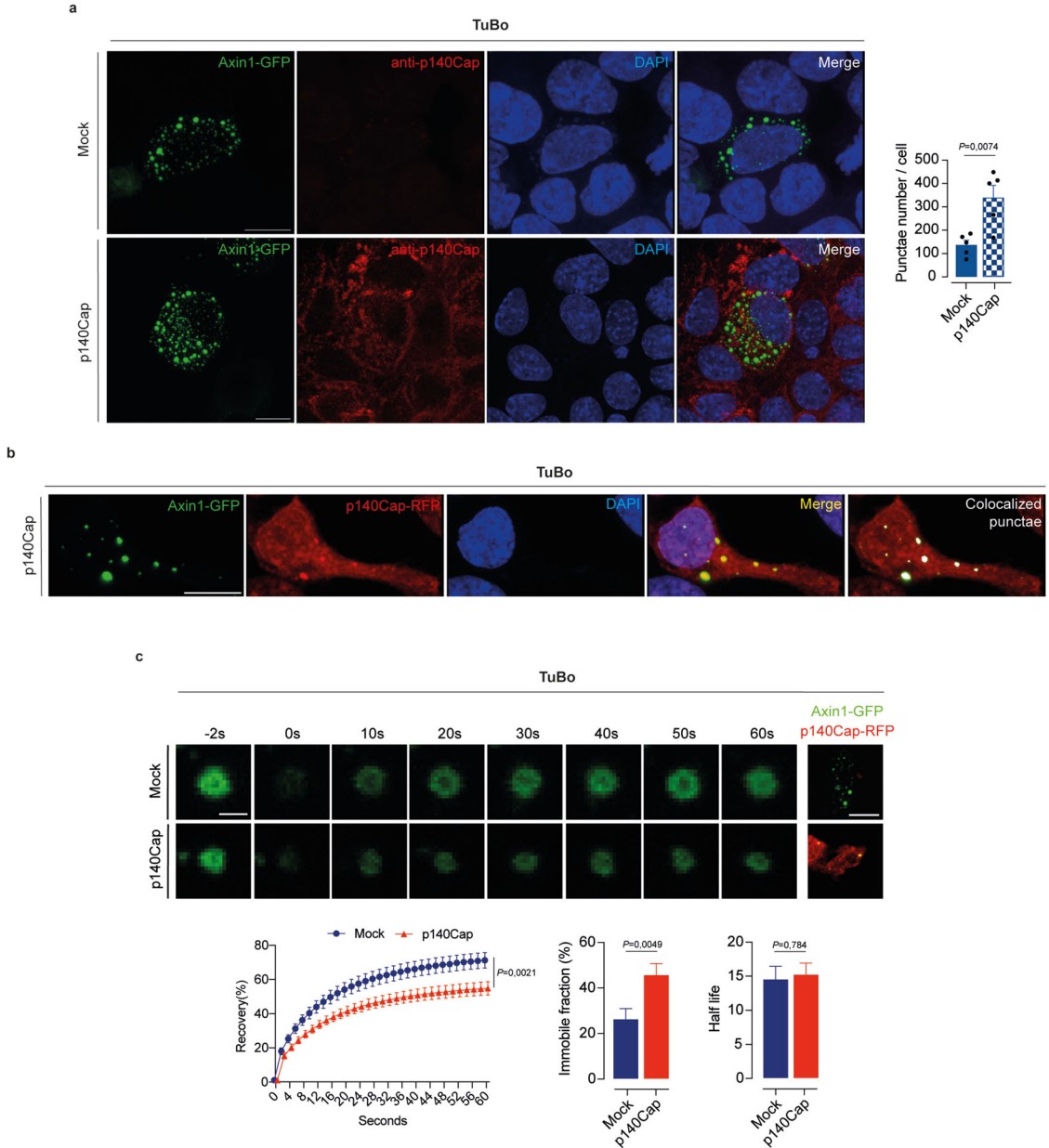

**Fig. 8 | p140Cap controls the stability of the destruction complex. a** Confocal images showing the number of Axin1-GFP punctae in Mock and p140Cap-TuBo cells. The nuclei were counterstained by DAPI. Scale bar 10 μm. Data are representative of *n* = 5 experimental repeats and shown as mean ± SEM; two-tailed unpaired *t* test. **b** Confocal images showing the colocalization of Axin1-GFP and p140Cap-RFP in punctae in Mock-TuBo cells co-transfected with Axin1-GFP and p140Cap-RFP. Scale bar 10 μm. Data are representative of *n* = 5 experimental repeats and shown as mean ± SEM; two-tailed unpaired *t* test. **c** Selected frames of FRAP experiments before and after bleaching, and throughout the recovery of Axin1-GFP in TuBo cells transfected with Axin1-GFP alone or in combination with p140Cap-RFP. Immobile fraction and half-life of GFP-Axin1. Scale bar 1 μm and 10 μm. Cell number > 31, data are representative of *n* = 3 experimental repeats and shown as mean ± SEM; two-tailed unpaired *t* test; 2way ANOVA for FRAP.

BC, providing substantial evidence that tumor-autonomous mechanisms of tumor suppression, such as the restriction of the TIC compartment through β-Catenin inhibition, are inextricably intertwined with local and systemic tumor-extrinsic effects that antagonize the establishment of tumor immune tolerance and inhibit tumor progression and metastasis. Our findings also provide formal demonstration of the role of p140Cap in directly inhibiting TICs through inhibition of β-Catenin, harnessing this function to anti-tumor regulation of the TME immune response (Fig. 10). In conclusion, our studies pave the way for the development of therapeutic intervention to target the epistatic β-Catenin/TIC/G-CSF/PMN-MDSCs axis to restore an efficient anti-tumor immune response, and point to the

clinical validity of p140Cap as a biomarker for prognostic patient stratification and for anti-Wnt/β-Catenin targeted therapy response prediction.

## Methods

Our research complies with all relevant ethical regulations following the rules of the Ethical Committee of our Institutions. For in vivo mouse experiments, only female mice were used, sample sizes were selected based on past experience and published experiments, and experiments included a minimum of three mice per group. Mice were housed in an animal facility with 12 h light/ 12 h dark cycle at 22 °C and 50% humidity with unrestricted access to food and water. Mice were

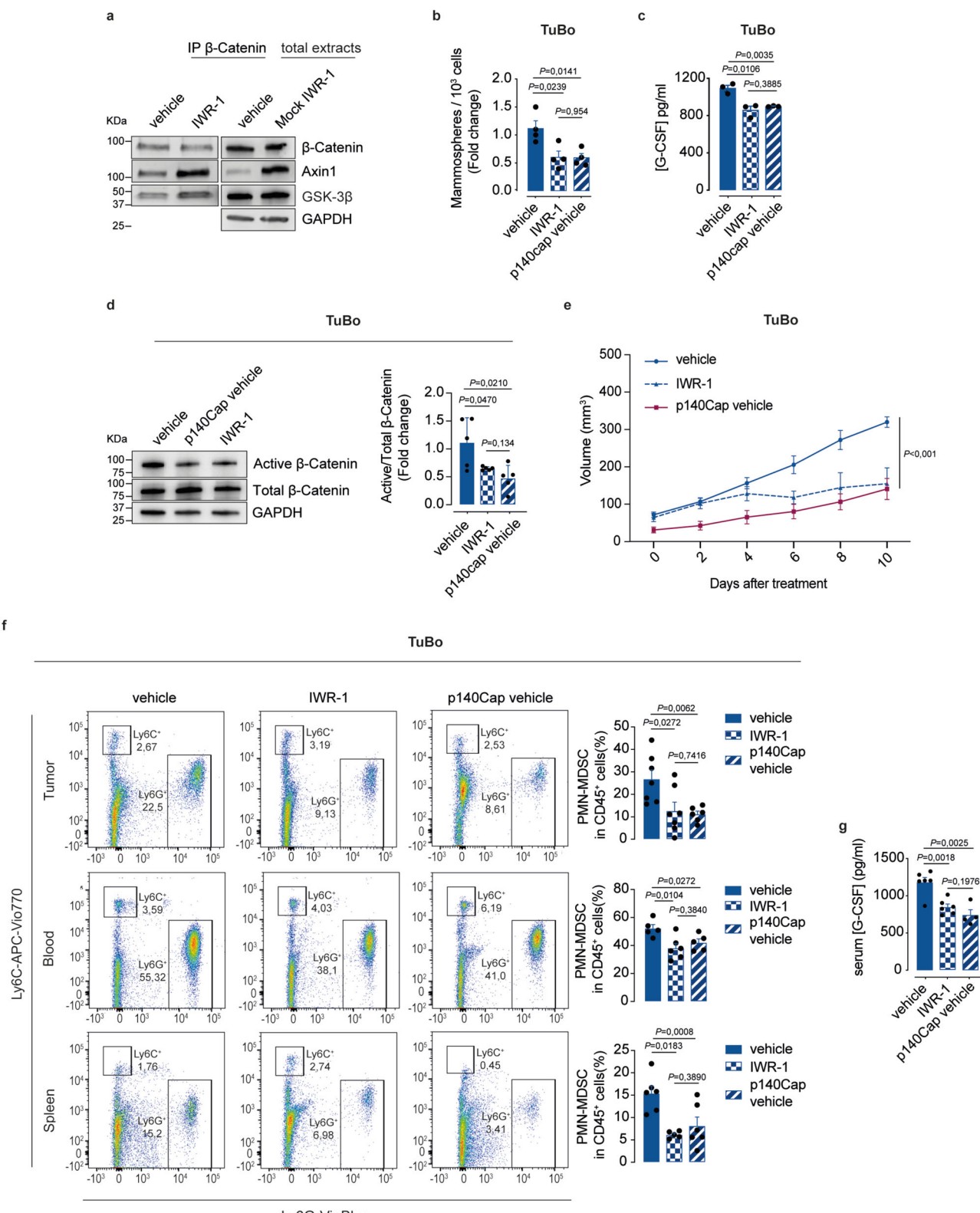

randomly assigned to each experimental group. The number of biological and experimental replicates is described in the figure legends. The protocol number from the Ministry of Health is CC652,72.

## Clinical cohorts and histopathological evaluation
A tissue microarray comprising a retrospective consecutive cohort of 622 female patients, from the European Institute of Oncology in Milan (IEO cohort), with complete clinicopathological follow-up, was used.

All patients provided written informed consent and underwent surgical procedures. Patients were stratified according to the immunohistochemical expression of p140Cap in the primary tumor according to an intensity score from 0 to 3 (0–0.5+, negative; 1+, weak positive; 2+, moderate positive; and 3+, strong positive), and stratified as p140Cap$^{LOW}$ (<1) or p140Cap$^{HIGH}$ (≥1), as previously described[23]. The density of TILs was assessed on hematoxylin and eosin (H&E)-fixed samples by detecting the total quantity of mononuclear immune cells

**Fig. 9 | IWR-1 Axin1 stabilizer mimics p140Cap ability to improve the destruction complex stability and to curb tumor progression. a** Effect of IWR-1 on the destruction complex stability. Immunoprecipitation of β-Catenin from Mock TuBo mammospheres treated for 5 consecutive days with 1 μM IWR-1 or DMSO (control vehicle). Immunoblot with indicated antibodies. GAPDH was used as loading control ($n = 3$). Data are representative of $n = 3$ experimental repeats. **b** Effect of IWR-1 treatment on mammosphere formation efficiency. Mammosphere formation assay from untreated Mock and p140Cap cells or Mock TuBo mammosphere treated with IWR-1 as in (**a**). Bar plots of mammosphere numbers are shown. Data are represented for $n = 4$ experimental repeats/group, as mean ± SEM; two-tailed unpaired $t$ test. **c** G-CSF protein levels in mammosphere supernatants, treated as in (**a**). G-CSF protein levels were measured by ELISA in supernatants from untreated Mock and p140Cap or Mock TuBo mammospheres treated with IWR-1 as in (**a**). Data are represented for $n = 3$ experimental repeats/group, as mean ± SEM; two-tailed unpaired $t$ test. **d** Immunoblot analysis of active β-Catenin in untreated Mock and p140Cap cells, or Mock TuBo mammospheres treated with IWR-1 as in (**a**). GAPDH was used as loading control. Data are representative of $n = 5$

experimental repeats and shown as mean ± SEM; two-tailed unpaired $t$ test. **e** $10^5$ cells were injected into the mammary fat pad of 6-weeks-old BALB/C mice, tumor growth was monitored and tumor size was measured. Mock TuBo tumor-bearing mice were treated with IWR-1 ($n = 5$) (5 mg/Kg) every two days, with six different injections, starting when the tumor size had reached approximately 80 mm³. Mock ($n = 6$) and p140Cap TuBo ($n = 6$) tumor-bearing mice were treated with DMSO as controls. Data are represented for $n = $ x mice and shown as mean ± SEM; 2way ANOVA. **f** Representative flow cytometry bar plots show the percentage of PMN-MDSCs cells normalized on CD45⁺ cells, in tumor, blood and spleen of mice described in (**e**). TuBo vehicle ($n = 7$/tumor, $n = 5$/blood, $n = 6$/spleen); TuBo p140Cap vehicle ($n = 6$/tumor, $n = 5$/blood, $n = 5$/spleen); TuBo IWR-1 ($n = 7$/tumor, $n = 6$/blood, $n = 6$/spleen). Data are represented for $n = $ x mice and shown as mean ± SEM; two-tailed unpaired $t$ test. **g** G-CSF levels in sera of IWR-treated or untreated tumor-bearing mice as in (**e**), by ELISA. TuBo vehicle ($n = 6$); TuBo p140Cap vehicle ($n = 4$); TuBo IWR-1 ($n = 6$). Data are represented for $n = $ x mice and shown as mean ± SEM; two-tailed unpaired $t$ test.

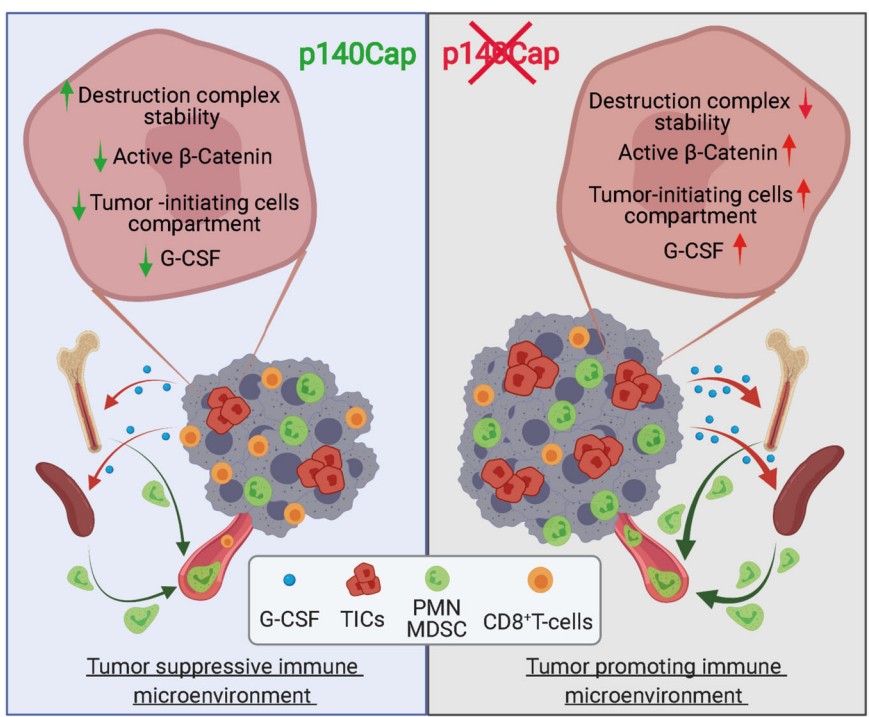

**Fig. 10 | p140Cap exerts a cell-extrinsic influence on the immune infiltrate composition of the tumor microenvironment, thereby favoring an effective anti-tumor immune response at the expense of a tumor-promoting inflammatory microenvironment.** Overall, this study shows that p140Cap activity is able to prevent tumor microenvironment infiltration through immunosuppressive PMN-MDSCs by inhibiting their systemic mobilization and local infiltration. Our

functional studies in vitro and in vivo revealed that the decrease in G-CSF content is a function of the p140Cap ability to reduce the TIC compartment, which in turn is responsible for the production of G-CSF. The effect on the TIC compartment depends on the ability of p140Cap to enter and stabilize the β-Catenin destruction machinery, thereby inhibiting β-Catenin activity. Created with Biorender.com.

---

as a function of the overall percentage of the stromal area within the borders of the invasive tumor. TILs in the tumor area with crush artifact, necrosis, or hyalinization were excluded. Polymorphonuclear leukocytes were also excluded. Areas of necrosis and massive inflammation as well as areas of lymphocytic infiltrate around normal lobules or previous biopsy sites were excluded from the analysis, following the international consensus guidelines for the histo-pathological assessment of TILs[31]. Data on TILs were available only for a total of 390 patients, due to the excessive compactness of tumor nests, with presence of virtual, not quantifiable stromal areas, and/or absence of tumor borders in the remaining cases. The patients were assigned to the TILs-positive group if their stromal TILs percentage was >1, and to the TILs-negative group if their stromal TILs percentage was 0.

## Antibodies and cell lines

Mouse monoclonal antibodies to p140Cap were produced at the Antibody production facility of the Department of Molecular Biotechnology and Health Sciences, University of Torino, by mice immunization with a recombinant bacterial p140Cap protein, obtained by fusing the sequence corresponding to amino acids 800–1000 of mouse SRCIN1 gene to the gene of the Glutathione S-transferase (GST). The resulting purified monoclonal antibodies were characterized by western blotting and IHC as shown in Supplementary Fig. 1 of[23]. For western blot analysis, the following antibodies were used: Tubulin (from the antibody production facility of the Dept of Molecular Biotechnology and Health Sciences, University of Turin), GAPDH (Millipore; Cat#3291346 1:8000); β-Catenin (BD Transduction Laboratories; Cat#610153; 1:1000); (Non-Phospho) Active β-Catenin

(Cell Signaling Technologies; Cat#8814 S; 1:1000), Axin1 (Cell Signaling Technologies; Cat#2087 S 1:1000), GSK-3β (Cell Signaling Technologies; Cat#9315 S; 1:1000), pGSK-3β (Cell Signaling Technologies; Cat#5558 S; 1:1000), Ubiquitin (Santa Cruz Biotechnology; Cat# sc-8017; 1:1000), Phospho β-Catenin (Cell Signaling Technologies; Cat# 9561 T; 1:1000), Oct-4 (Cell Signaling Technologies; Cat#75463 S; 1:1000), mouse IgG (Santa Cruz Biotechnology; Cat#sc-2025), rabbit IgG (Santa Cruz Biotechnology; Cat#sc-2025). Secondary antibodies conjugated with peroxidase were purchased from GE Healthcare (goat anti-mouse IgG Cat#31430; goat anti-rabbit IgG Cat#31460). Nitrocellulose, and films were obtained from GE Healthcare (Buckinghamshire, UK).

Mycoplasma-free TuBo cells derived from a spontaneous breast tumor arisen from BALB/c-MMTV-NeuT mice[35]. 4T1 (ATCC CRL-2539) and MDA-MB-231 (ATCC HTB-26) cells were purchased from ATCC (LGC Standards S.r.l.-Italy Office, Italy), authenticated and tested for the absence of mycoplasma. TuBo cells were cultured in DMEM medium, supplemented with 20% FBS. 4T1 cells were cultured in RPMI medium, supplemented with 10% FBS. MDA-MB-231 cells were cultured in DMEM medium, supplemented with 10% FBS. Culture media and FBS were from Invitrogen (Carlsbad, CA, USA). HEK293 cells were cultured in DMEM medium, supplemented with 10% FBS. All culture media were supplemented with 1% Pen/Strep from Invitrogen (Carlsbad, CA, USA).

## Cell lysis, immunoblotting and immunoprecipitation

Cells were extracted using a RIPA buffer (50 mM Tris (pH7.5), 150 mM NaCl, 1% Triton X100, 1% Na Deoxycolate, 0.1% SDS and protease inhibitors). Cell lysates were centrifuged 20 min at 16,300 x g, and the supernatants were collected and assayed for protein concentration using the Bio-Rad protein assay method (Biorad, Hercules, CA, USA). Proteins were run on SDS−PAGE under reducing conditions. Following SDS−PAGE, proteins were transferred to nitrocellulose membranes, saturated with 5% BSA for 2 h, incubated with specific antibodies and then detected with peroxidase-conjugated secondary antibodies and the chemiluminescent ECL reagent (Biorad). For immunoprecipitation experiments, proteins from TuBo and 4T1 cells were extracted with Lysis buffer (150 mM NaCl, 50 mM Tris pH 7.4, 1% NP-40, 1 mM MgCl2. 5% glycerol), for 1 h at 4 °C. One or 2 mg of proteins were immunoprecipitated with 1 or 2 μg of antibodies to p140Cap, β-Catenin, Axin1, GSK-3β, for 2 h at 4 °C in the presence of 7 or 14 μl of protein G Dynabeads (Invitrogen, Carlsbad, CA, United States). Beads were washed seven times with cold lysis buffer, resuspended in 25 μl of 2% SDS-PAGE sample buffer in reducing conditions, and incubated at 95 °C for 10 min. Following SDS−PAGE, proteins were transferred to Nitrocellulose membranes, and analyzed as shown above.

## Retrovirus production and cell infection

To express p140Cap into TuBo, MDA-MB-231 and 4T1 cells, p140Cap cDNA was cloned into pBabe-puro. The retrovirus particles were produced by exploiting the Lipo2000 (ThermoFisher Scientific) transfection of Platinum Retroviral Packaging Cell Lines (Cell BioLabs), in 6 mm dishes. 48 h after transfection, supernatants that contained the retrovirus particles were collected and added directly to subconfluent cells, cultured in 6-well plates. After 48 h, cells were washed and cultured with a selection medium containing puromycin (Sigma-Aldrich) at a final concentration of 1 mg/ml. The efficiency of infection was controlled by western blot and immunofluorescence analysis. Cells were then cultured at a single cell level in four 96-well adherent multiwell plates (Corning). Individual p140Cap expressing clones were isolated 20 days after the start of the selection, by assessing the p140Cap expression through western blot and immunofluorescence. Four individual positive clones were pooled together to rule out clonal artifacts.

## DNA constructs

The p140Cap BirA: MCS-BirA(R118G)-HA and pcDNA3.1mycBioID-p140Cap, BirA: MCS-BirA (R118G)-HA, pcDNA3.1mycBioID plasmids were obtained from Professor Thilo Kähne (Otto-Von-Guericke, University of Magdeburg). The AXIN1-GFP: pRP[Exp]-CMV > hAXIN1[NM_003502.4]/EGFP (VectorBuilder Inc.)

## RNA extraction, retrotranscription and RT-PCR

Total RNAs were extracted using TRIzol reagent (Invitrogen, Carlsbad, CA; Cat#15596018) according to the manufacturer's instructions. RNA was re-suspended in sterile pure water and stored at −80 °C. Before RNA retro-transcription and RT-PCR, samples were gently thawed on ice. Integrity and purity of RNA were assessed using Nanodrop ND-100 Spectrophotometer (Nanodrop Technologies, Wilmington, USA). For RT-PCR, 1 μg of total RNA was used for cDNA synthesis with random hexamers. RT-PCR was carried out using a 7300 Real Time PCR System (Applied Biosystems). Reactions were run in triplicate in three independent experiments. The geometric mean of housekeeping gene 18 S was used as an internal control to normalize the variability in expression levels. Expression data were normalized to the geometric mean of housekeeping gene 18 S to control the variability in expression levels and were analyzed using the 2-ΔΔCT method. The following primers were used for qPCR:

-Mouse G-CSF: GTTGTGTGCCACCTACAAGC (Forward 5′-3′), CCATCTGCTGCCAGATGGTGGT (Reverse 5′-3′);

-Human G-CSF: GGAGAAGCTGGTGAGTGAGTGT (Forward 5′-3′), CCAGAGAGTGTCCGAGCAG (Reverse 5′-3′).

## RNA-Seq analysis

For RNA-Seq analysis, RNAs were quantified using a Nanodrop, as above, and a 2100 Bioanalyzer (Agilent RNA 6000 Nano Kit, Waldbronn, Germany); RNAs with a 260:280 ratio of ≥1.5 and an RNA integrity number of ≥8 was deep sequenced. Sequencing libraries were prepared with the Illumina TruSeq Stranded RNA Library Prep, version 2, Protocol D, using 500-ng total RNA (Illumina, USA). The qualities of the libraries were assessed by 2100 Bioanalyzer, with a DNA1000 assay. Libraries were quantified by RT-PCR using the KAPA Library Quantification kit for Illumina sequencing platforms (KAPA Biosystems); RNA processing has been carried out using Illumina NextSeq 500 Sequencing, using 6 samples for each run, mixing samples and controls in each flow cell, to avoid not manageable batch-effects. FastQ files were generated via Illumina bcl2fastq2 (Version 2.17.1.14) starting from.bcl files produced by Illumina NextSeq sequencer. All bioinformatics analyses were performed using tools compliant with data reproducibility[78], and implemented in docker4seq[79] and rCASC[80].

## ELISA assay

Supernatants from TuBo, 4T1 and MDA-MB-231 cells were harvested after 3 days of culture. The final G-CSF concentration was normalized on the number of cells and expressed as pg/ml/10^6 cells. For mammospheres, supernatants were collected after 5 min centrifugation at 4000 × g. For serum, blood collected by cardiac puncture was processed 10 min at 37 °C. To separate serum from blood cells, the samples were centrifuged 10 min at 5000 × g. Plasma and cell culture supernatants were stored at −80 °C before G-CSF analysis by murine and human standard ABTS ELISA Development Kit (Peprotech; Cat#900-K103; Cat#900-K77) according to manufacturer's instructions. Data were obtained with the GloMax Discover Microplate Reader (Promega).

## In vivo tumor growth and metastasis assessment

The in vivo studies were performed only in female mice. Six/eight week-old female BALB/c mice were purchased from Charles River Laboratories (Calco, Italy) and treated in accordance with the European Community guidelines. The study received ethical approval

from the Ministry of Health, protocol number CC652.72. $10^4$ and $10^5$ 4T1 and TuBo cells, respectively, were resuspended in 50 μl of PBS and then injected into the left fat pad of BALB/c mice. The size of the tumors was evaluated every two days using a digital caliper in blind experiments: maximum and minimum diameter were measured and the volume was calculated using the following ellipsoid formula: $(4/3\pi(d/2)^2*D/2)$. Mice were euthanized using a $CO_2$ chamber when the tumor was approximately 500 mm³. This maximum tumor size, approved by our study protocol, was not exceeded. During the experiments, the tumor-bearing mice did not undergo more than 10% of weight loss. For the metastatic burden assessment, lungs of 4T1 tumor-bearing mice were fixed in 10% neutral buffered formalin (BioOptica) and paraffin-embedded. To optimize the detection of microscopic metastases and ensure systematic uniform and random sampling, lungs were cut transversely, to the trachea, into 2 mm-thick parallel slabs with a random position of the first cut in first 2 mm of the lung, resulting in 5–8 slabs for lung. The slabs were then embedded cut surface down and sections were stained with hematoxylin and eosin (BioOptica). The metastatic lung tissue was evaluated with Adobe Photoshop by selecting metastases with the lasso tool and reporting the number of pixels indicated in the histogram window as percentage of the total lung area.

## Mouse-derived sample processing and FACS analysis

For FACS analysis of immune cells, tumor, blood, spleen and bone marrow were collected from BALB/c tumor-bearing mice, after the injection of TuBo or 4T1 cells in the mammary fat pad. Tumors of the same size were always compared. Tumors were placed in a 50 ml tube with 5 ml of cold serum-free DMEM, while spleen and bone marrow in 5 ml of cold PBS. Tumors were processed as follows: samples were mechanically dissociated using a scalpel, until a well-disaggregated pulp was formed. The tissue pulp was re suspended in 5 ml DMEM serum-free supplemented with 0.1 mg/ml Collagenase (Sigma-Aldrich; Cat#C0130) and incubated at 37 °C degrees, under rotation, for 1 h. The digested tissue pulp was filtered using a 40 μm cell-strainer (Falcon; Cat#352340) in order to obtain a single cell suspension; samples were then centrifuged 5 min at 476 g at 4 °C; supernatant was removed and samples were washed using cold PBS and stored on ice. Spleens were mechanically dissociated using the back of a 2.5 ml syringe and cold and sterile PBS within a 40 μm cell strainer in order to obtain a single cell solution; the sample was centrifuged 5 min at 476 g at 4 °C; supernatant was removed and samples were washed using cold PBS and stored on ice until further steps. Bone marrows were extracted from both the femoral bones of mice and processed as follows: femoral bones were carefully cut at the extremities to maintain the bone integrity. A 26-gauge needle and a 1 ml syringe filled with cold serum-free DMEM were used to flush the bone marrow out onto a 70 μm nylon cell strainer placed in a 6 cm diameter petri dish. The back of a 2.5 ml syringe was used to disaggregate bone marrow clusters and obtain a single cell suspension in DMEM; the suspension was pipetted into a 50 ml plastic tube and centrifuged 5 min at 476 g at 4 °C; supernatant was removed and the samples were washed using cold PBS in appropriate volume according to the pellet dimension and store on ice until further steps. Blood samples were collected in 1.2 ml, K3 EDTA S-MONOVETTE tubes (SARSTEDT; Cat#1664001) to avoid coagulation.

Once tumor, spleen, blood and bone marrow samples were ready, they were dispensed in FACS tubes. Red blood cell lysis was performed by adding 500 ml of red blood cells lysis solution in each tube and incubated 10 min at room temperature. After this step, all samples were centrifuged 5 min at 476 g at 4 °C and washed with cold PBS. Excess of supernatant after washing was eliminated by flicking the tube rapidly. To rule out staining artifacts, a mix of each FACS antibody was prepared and dispensed in appropriate volume in each FACS tube, followed by a 30 min incubation at 4 °C in the dark. Unstained and FMO controls were used to regulate physical and electrical parameters at the FACS machine and to design the gating strategy. The following antibodies were used (at a dilution of 1:200): CD45-VioGreen (Cat#130-110-803), CD11b-FITC (Cat#130-110-803), Ly6G-VioBlue (Cat#130-119-986), Ly6C-APC-Vio770 (Cat#130-121-439), F4/80-PE-Vio770 (Cat#130-118-320), MHC-II-APC (Cat#130-102-139), CD3-FITC (Cat#130-119-135), CD4-APC-Vio770 (Cat#130-119-134), CD8-VioBlue (Cat#130-123-865), CD49b-PE (Cat#130-123-702) all from Miltenyi Biotec (Miltenyi Biotec B.V. & Co. Bologna, Italy), while CD206-PE (Cat#141706) was from BioLegend (BioLegend, San Diego CA, USA).

For FACS analysis, a BD FACSVerse machine was used and 510000 CD45$^+$ events for each sample were acquired. The software used for acquisition and analysis of data was BD FACSSuite. To assess the levels of all the immune cell population reported in this study and reach statistical significance, at least from five to eight independent experiments were performed.

Cells were gated according to their physical parameters, and dead cells were excluded following staining with propidium iodide (Sigma-Aldrich; Cat#P4864) and CD45$^+$ leucocytes were gated. The percentage of CD11b$^+$-Ly6G$^+$-PMN-MDSC, CD11b$^+$-Ly6C$^+$-M-MDSC, CD11b$^+$-F480$^+$-MHC-II$^+$-M1 and CD11b$^+$-F480$^+$-CD206$^+$-M2 macrophages, of CD3$^+$-CD4$^+$- or CD8$^+$-T lymphocytes and of CD3–CD49b$^+$- Natural Killer cells on total CD45$^+$-leucocytes was analyzed; see Supplementary Fig. 8a for the gating strategy

To assess the percentage of cancer stem cells in TuBo, 4T1 and MDA-MB-231, cells were washed with sterile PBS and detached using Trypsin. Trypsin was then inactivated with DMEM complete medium and cells were centrifuged 5 min at 476 g at 4 °C. Supernatant was removed and cells were resuspended in an appropriate volume and counted. A total amount of 1 million cells were then put into a FACS tube, and incubated with antibodies for 30 min at 4 °C in the dark. For the FACS staining, we used CD44-PE (Cat#103007), CD24-PE/Cy7 (Cat#101822) and Sca-1-AlexaFluor 647 (Cat#122518) antibodies, all from Biolegend, diluted at 1:200. A total of 10,000 live CD45- cells/events were set for acquisition and data analysis; for the gating strategy see Supplementary Fig. 8b.

## Mammosphere formation assay

TuBo and 4T1 cells from cultures or dissociated tumors (see below), were plated at a density of $6 \times 10^4$ cells/ml in ultra-low attachment 10-cm plates (Corning) in mammosphere serum-free DMEM-F12 medium (Invitrogen), supplemented with 20 ng/ml basic Fibroblast growth factor (bFGF; Peprotech; Cat#100-18B), 20 ng/ml Epidermal growth factor (EGF; Sigma-Aldrich; Cat#E9644), 5 μg/ml Insulin (Sigma-Aldrich; Cat#I9278), and 0.4% Bovine Serum Albumin (BSA, Sigma). MDA-MB-231 cells were detached and plated at a density of $6 \times 10^4$ cells/ml in ultra-low attachment 60-mm plates (Corning) in mammosphere medium, serum-free DMEM-F12 medium (Invitrogen) supplemented with 2% B27 (Invitrogen; Cat#17504-044;), 20 ng/ml EGF, 0.4% BSA and 4 μg/ml Insulin. All the culture media were supplemented with 1% Pen/Strep (Invitrogen) antibiotics. Non-adherent spherical clusters of cells were collected after 5 days. Spheres were counted by plating 25 μl of sphere suspension in four well of 96-well plate (Corning) and the total number of spheres in all wells (100 μl) was derived after counting under optical microscope. The number of mammospheres was indicated as mammosphere number generated from every $10^3$ single cells plated, using the following formula: X(spheres/$10^3$ cells) = [(total number of spheres*total volume of culture medium*$10^3$)/volume of sphere count]/initial number of plated cells.

For the evaluation of mammosphere, at least 10 images of sphere culture were collected using a Zeiss microscopy (Oberkochen, Germany) at ×4 magnitude after 5 days of culture.

## In vivo treatments: anti-Ly6G Mab or IWR-1

$10^5$ TuBo cells were injected orthotopically in the fourth mammary gland on the left side of BALB/c mice. Treatments started when tumor size reached an average of 80 mm³. Anti-Ly6G treatments consisted of two IP injections per week of 100 µg anti-Ly6G (BioXCell; Cat#BE0075-1) or IgG2a isotype control (BioXCell; Cat#BE0089) in PBS. IWR-1 (Sigma; Cat#I0161) treatment consisted of six different intratumoral injections every two days using a dose of 5 mg/Kg in DMSO.

## Generation of the constitutive active β-Catenin cells

Mutant β-Catenin pcDNA3-S33Y was a kind gift from Eric Fearon (Addgene plasmid #19286), pcDNA™3.1(+)/mycHis A (Cat#2094) was purchased from Invitrogen (Invitrogen, Carlsbad, CA). Stable transfection was performed using the Lipofectamine® 2000 DNA Transfection Reagent Protocol (Thermo Scientific, Whaltman, MA, USA) according to manufacturer's instructions. In particular, pcDNA3-S33Y vector was used to transfect TuBo-p140Cap cells, while pcDNATM 3.1/myc-His A vector for both TuBo-p140Cap and -Mock cells as controls. G418® (80 µg/ml) antibiotic treatment (Thermo Scientific, Whaltman, MA, USA) was used as a positive selection for cells expressing the neomycin resistance (neo) gene. The resistant cell colonies were isolated after 14 days of culture in presence of G418 selection by picking them from the starting plate and by transferring them in a 96-well plate to permit the growth of the single clones. Positive clones were screened by western blot and immunofluorescence analyses for the active β-catenin expression. Five individual positive clones were pooled together to rule out clonal artifacts.

## BioID assay

HEK293 cells were transiently transfected with Lipofectamine 2000 (Invitrogen, USA) according to manufacturer's protocol. The day after the transfection the cells were treated for 3 h with D-biotin 50 µM (Thermofisher Cat# B20656) diluted in the culture medium. Cells were incubated twice with biotin-free medium for 1 h, lysed with 500 µl Buffer 2 (150 mM NaCl; 20 mM TRIS-HCl pH 7,5; 5 mM EDTA; 12 mM Deoxycholate sodium; water), scraped and incubated for 15 min at 4 °C. Lysed cells were collected, cropped with 2 6 G injection needle, sonicated (4 hits for 15 s, 30% power) and centrifuged at 4 °C (15,000 g) for 15 min. Supernatants were collected and quantified. 2 mg of extracts were incubated for 2 h with 40 µl of streptavidin-conjugated resin (Thermofisher High-Capacity Streptavidin Agarose Resin Cat# 20357). After 2 h the resins were centrifuged at 4 °C (2500 g) for two minutes and the supernatant was removed. Five washes of 1 ml were performed alternating resuspension and centrifugation: one wash in Buffer 1 (SDS 2% in deionized water); two washes in Buffer 2 (previously described); one wash in Buffer 3 (150 mM NaCl; 20 mM TRIS-HCl pH 7,5; 5 mM EDTA; 12 mM Deoxycholate sodium; 1% NP40; 0,01% SDS; 1% Triton X-100; deionized water); one wash in Buffer 4 (150 mM NaCl; 20 mM TRIS-HCl pH 7,5; 5 mM EDTA; 12 mM Deoxycholate sodium; 1% NP40; 0,01% SDS; 0,01% Triton X-100; deionized water). After last washing supernatant was removed with a Hamilton needle. Biotinylated proteins were detached from the resin adding a volume of reduced Sample Buffer (8% SDS) with an excess of D-biotin (1 mM) and heating them at 95 °C for 15 min. Eluted proteins were finally collected with a Hamilton needle.

## Immunofluorescence

For immunostaining, Mock and p140Cap-TuBo and COS-7 cells were transfected with Axin1-GFP alone or in combination with p140Cap-RFP were fixed at RT with 2% paraformaldehyde (PFA) for 10 min. Cells were then permeabilized and saturated with 0.1% Triton-X-100, 1% BSA in PBS for 10 min. For the anti-p140Cap immunofluorescence, Mock and p140Cap-TuBo cells were incubated with primary antibody p140Cap (1:500 at RT for 60 min. Primary antibodies were detected with anti-mouse Alexa Fluor 568 (Molecular Probes, Invitrogen), used at 1:500

dilution for 30 min. Nuclei staining was performed with the DNA dye DAPI (Sigma) at 0.5 µg/ml in PBS for 10 min at RT and mounting was performed with ProLong (Invitrogen; Cat# P36930). Images were acquired using a Leica SP8 confocal system with HyVolution 2 (Leica Microsystems) equipped with an argon ion, a 561 nm DPSS and a HeNe 633 nm lasers. Fixed cells were imaged using a HCX PL APO 63×/1.4 NA oil immersion objective. Series of x-y-z images were collected. All the images were analyzed by using Fiji Software (an image processing package distribution of ImageJ, USA).

## FRAP analysis

Mock-TuBo cells were transfected with Axin1-GFP alone or in combination with p140Cap-RFP. Imaging was performed using a Leica TCS SP5 confocal system (Leica Microsystems) equipped with an argon ion and a 561 nm DPSS lasers. Cells were imaged using a HCX PL APO 63×/1.4 NA oil immersion objective.

For FRAP experiments, cells were kept in the microscope incubator at 37 °C and 5% CO2. Photobleaching was performed on GFP-Axin1 punctae of constant size using 100% transmission of 488 and 568 nm lasers. Recovery kinetics was then measured and, after background subtraction, were normalized to pre-bleach frames.

## Ubiquitination assay

Mock and p140Cap-TuBo mammospheres were treated with 10uM of MG132 (Millipore; Cat# 474790) for 8 h. Proteins from TuBo mammospheres were extracted with Lysis buffer (150 mM NaCl, 50 mM Tris pH 7.4, 1% NP-40, 1 mM MgCl2, 5% glycerol and 10 mM Iodoacetamide), for 1 h at 4 °C. 2 mg of proteins were immunoprecipitated with 2 ug of antibodies to p140Cap for 2 h at 4 °C in the presence of 14 ul of protein G Dynabeads (Invitrogen, Carlsbad, CA, United States). Beads were washed ten times with cold lysis buffer, resuspended in 25 µl of 2% SDS-PAGE sample buffer in reducing conditions, and incubated at 95 °C for 10 min. Following SDS–PAGE, proteins were transferred to Nitrocellulose membranes, and analyzed as shown above with the anti-Ubiquitin antibody.

## Functional immune assays

Mouse Ly6G+ and CD11b+ cells were isolated from spleen and primary tumor of tumor-bearing mice, respectively, by immunomagnetic sorting (Miltenyi Biotec, Bologna, Italy) according to the manufacturer's instructions. Cells purity was evaluated by flow cytometry. The cell immunosuppressive activity was evaluated by plating Ly6G+ or CD11b + cells in co-culture with splenocytes from Thy-1.1 Balb-c transgenic mice expressing a Kd-restricted HA512-520 peptide-specific α/β TCR (CL4 mice, a gift from L. Sherman - The Scripps Research Institute, La Jolla, CA, USA) diluted 1:10 with Thy1.2 Balb-C (Charles River Wilmington, MA, USA) naïve splenocytes, in the presence of HA512-520 peptide (1 µg/ml, final concentration). Sorted cells were plated in 96 wells plate at a final concentration of 24% of total cells. The splenocytes of transgenic mice were labeled with 1 µM CellTrace (Thermo Fisher Scientific, Waltham, MA, USA). After 3 days of co-culture, cells were stained with APC-Cy7 conjugated anti-CD90.1 (clone OX-7, Biolegend San Diego, CA, USA) and PerCP-Cy5.5 conjugated anti-CD8 (clone SK1, eBioscience, Thermo Fisher Scientific, Waltham, MA, USA) mAb. CellTrace signal of gated lymphocytes was used to analyze cell proliferation and the results were reported as percentage of dividing CD8 + cells for each generation. Samples were acquired with FACS-Canto II (BD, Franklin Lakes, NJ, USA) and the data were analyzed by FlowJo software (Tree Star, Inc. Ashland, OR, USA).

## Gene set enrichment analysis (GSEA)

The TCGA[34] and CCLE[56] ranked lists were based on the Spearman correlation coefficient of the expression profile of each expressed gene with that of p140Cap. The TUBO ranked list was based on the expression logFC between the p140Cap expressing- and -mock cells.

The pre-ranked GSEA analysis[32,33] was performed using R package 'cluserprofiler' [https://www.R-project.org/][81] with default settings on MSgDB[82] v7 data sets.

## Statistical analysis

In the evaluation of the stromal TILs, the Pearson test was used to assess the relationship between stromal TILs and p140Cap levels (by IHC). Overall survival of the stromal TILs-negative and -positive groups was analyzed using the Kaplan-Meier survival analysis, and the statistical significance of between-group differences was evaluated using the Wald test. Univariate and multivariate Cox regression analyses were performed to identify the prognostic significance of stromal TILs.

Differences in the growth rate of mouse tumors were analyzed with two-way ANOVA followed by Bonferroni multiple comparison post hoc test. For quantification, statistical significant differences were evaluated using unpaired *t*-tests. Error bar: s.e.m., using the Student's *t*-test.

## Data availability

Source data are provided with this paper. The raw and processed RNA sequencing data generated in this study have been deposited in the NIH GEO repository under accession code GSE176229. RNA (assessed by RNA sequencing) and protein (assessed by mass spectrometry) expression data and copy number status (assessed by SNP arrays) of SRCIN1 for the TGCA cohort PanCancer Atlas[34] were retrieved from cBioportal. Human and mouse gene set annotations were retrieved from MSigDB with the msigdbr package (version 7.4.1) in R. Human GRCh38 and mouse GRCm39 reference genomes (primary genome assembly). The remaining data are available within the Article, Supplementary information or Source Data file. Source data are provided with this paper.

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

## Acknowledgements

We thank G. Jodice, F. Montani and F. Sanguedolce at the Molecular and Digital Pathology Unit of the European Institute of Oncology (IEO) for tissue processing and staining; D. Disalvatore, R. Bonfanti and S. Capoano for help with pathological evaluation and statistical analyses.

The research leading to these results has received funding from AIRC under IG 2017 – ID. 20107 project and IG 2022 - ID. 27353 project – P.I. Defilippi Paola. This work was also supported by: Fondazione CRT 2020.1798, RILO University of Torino, Ministero della Salute (RF-2021-12371961) to Paola Defilippi, and PNRR M4C2-Investimento 1.4-CN00000041 Financed from −"NextGenerationEU" to Paola Defilippi and Daniela Taverna, AIRC-IG 11904, IG 15538, and MultiUnit -5×1000 MCO 10.000 to Salvatore Pece, The Italian Ministry of University and Scientific Research (MIUR) PRIN 20177E9EPY_002 and MIUR-PRIN 202032AZT3_004 to Salvatore Pece, The Italian Ministry of Health with Ricerca Corrente, Fondazione Umberto Veronesi (FUV) and 5×1000 funds to Salvatore Pece.

## Author contributions

V.S., D.TOS., D.TAV., E.T., S.P. and P.D. designed the research, and analyzed the data; V.S. performed and supervised all the experiments; S.P., D.TOS., S.F., F.A.T., and G.B. analyzed the human BC samples; V.S., M.V., A.S., F.M., C.A., A.M., performed the in vivo experiments; L.C. and F.C. analyzed the immune TME; A.P., G.C., M.M., A.G., D.N. performed the mammosphere and the destruction complex experiments; A.P. and M.G. performed the FRAP analysis; A.L. and M.I. analyzed the metastasis; R.C., D.I., S.O. performed the RNA seq analysis; E.M., F.A.T. and P.P. performed the bioinformatics analysis; F.D.S., C.F., S.U., V.B. performed the immunosuppressive assays; V.S., D.TOS., D.TAV., E.T., S.P. and P.D. wrote the manuscript; S.P. and P.D. funding acquisition. All authors reviewed the manuscript.

## Competing interests

The authors declare no competing interests.

## Additional information

**Vincenzo Salemme** [1,2], **Mauro Vedelago** [1], **Alessandro Sarcinella**[1], **Federico Moietta** [1], **Alessio Piccolantonio** [1,2], **Enrico Moiso**[1], **Giorgia Centonze** [1,2], **Marta Manco** [1], **Andrea Guala**[1], **Alessia Lamolinara** [3], **Costanza Angelini**[1], **Alessandro Morellato**[1,2], **Dora Natalini** [1], **Raffaele Calogero** [1,2], **Danny Incarnato** [4], **Salvatore Oliviero** [2,5], **Laura Conti** [1,2], **Manuela Iezzi** [3], **Daniela Tosoni** [6], **Giovanni Bertalot**[6], **Stefano Freddi**[6], **Francesco A. Tucci**[6,7], **Francesco De Sanctis**[8], **Cristina Frusteri**[8], **Stefano Ugel** [8], **Vincenzo Bronte** [8,9], **Federica Cavallo** [1,2], **Paolo Provero** [10], **Marta Gai**[1], **Daniela Taverna** [1,2], **Emilia Turco**[1], **Salvatore Pece** [6,11] ✉ & **Paola Defilippi** [1,2] ✉

[1]Department of Molecular Biotechnology and Health Sciences, University of Torino, Via Nizza 52, 10126 Torino, Italy. [2]Molecular Biotechnology Center (MBC) "Guido Tarone", Via Nizza, 52, 10126 Turin, Italy. [3]Immuno-Oncology Laboratory, Center for Advanced Studies and Technology (CAST), Department of Neuroscience, Imaging and Clinical Sciences, G. d'Annunzio University of Chieti-Pescara, Chieti-Pescara, Italy. [4]Department of Molecular Genetics, Groningen Biomolecular Sciences and Biotechnology Institute (GBB), University of Groningen, Groningen, the Netherlands. [5]Department of Life Sciences and Systems Biology, University of Turin, Torino, Italy and IIGM, Candiolo, Italy. [6]European Institute of Oncology IRCCS, 20141 Milan, Italy. [7]School of Pathology, University of Milan, Milan, Italy. [8]Immunology Section, Department of Medicine, University of Verona, 37134 Verona, Italy. [9]Istituto Oncologico Veneto, IRCCS, 35128 Padova, Italy. [10]Neuroscience Department "Rita Levi Montalcini", University of Torino, Via Cherasco 15, 10126 Torino, Italy. [11]Department of Oncology and Hemato-Oncology, Università degli Studi di Milano, 20142 Milano, Italy. ✉e-mail: salvatore.pece@ieo.it; paola.defilippi@unito.it

