## [Peer Review File · Nature Communications]

Reviewers' Comments:

Reviewer #1:

Remarks to the Author:

This is a continuation of this group's studies of the role of the adaptor protein and putative tumor suppressor p140Cap in cancer primarily in neuroblastoma and more recently breast cancer. This study focuses on its role in TICs, how it might regulate the β -catenin destruction complex and the tumor immune microenvironment. Previous studies have suggested a role for p140CAP through its interaction with p130Cas on Src activation. The authors use two "syngeneic" Balb/c models, the well studied 4T1 model and TuBo cells from MMTV-NeuT mice. Since NeuT is a rat gene this model may not be totally "syngeneic" and this may partially impact their immune profiling. My main concern with this study is the lack of mechanistic insight about how p140Cap regulates the destruction complex and how this leads to an increase in G-CSF. These two observations are the real novelty of this study. The other observations on correlating TILs with patient outcome, G-CSF regulating MDSCs, β -catenin regulating TICs are quite predictable from earlier studies. The actual effect on tumor growth of genetically manipulating p140Cap levels (Fig. 2A) are modest, and the treatment response with IWR-1 (Fig. 8E) is not that impressive. I am not convinced that "basally low p140Cap expression will be a useful biomarker to stratify BC patients", and that "drugs able to stabilize the β -catenin destruction complex will improve therapeutic response". The blood levels of PMN-MDSCs may provide a better and less invasive alternative than measuring p140Cap. Drugs regulating β -catenin have shown little utility to date, and have potential toxicities. A number of other minor concerns are listed below:

Page 3, Line 3 – Update estimated cases for 2022.

Page 6, Line 6 – What about triple negative breast cancers? TNBCs also have a high risk of metastatic recurrence. What are the levels of p140Cap in TNBCs?

Page 6, Line 14- Authors need to do additional experiments to identify TILs, either by IHC staining of CD3+ and CD20+ or Immunofluorescence or multiplex IF.

Page 7, Lines 2 – If a correlation exists between TILs and favorable disease outcomes in HER2+ patients, then what are the levels of p140Cap in these patients (aside from correlation analysis)? Have these patients received neo-adjuvant therapy? Why are they infiltrated by lymphocytes?

Page 7, Line 7 – Is the TCGA dataset of 1095 transcripts from HER2+ patients, TNBC, ER+? If the authors have these data they can study the transcript levels of SRCIN1 within each subtype.

Page 8 – Figure 2A, the authors should display images of H&E stained lung tissues to help confirm these findings.

Page 8 – What markers were utilized to distinguish between M1 and M2 like macrophages?

Page 13 – Why use MDA-MB-231 since the manuscript was focused on HER2+ BCs?

Reviewer #2:

Remarks to the Author:

The authors demonstrate using transcriptomics and preclinical mouse models that p140Cap expression correlates with enhanced presence of TIL and counteraction of MDSCs and can be considered a novel tumor suppressor. Specifically decreased G-CSF via p140Caps stabilizes of beta-catenin in the tumor initiating cell compartment/ tumor stem cells, which leads to prevention of MDSC infiltration and as a result decreased suppression of the TME. The authors also propose p140Cap could be a biomarker to help stratify tailored therapy. Specifically suggesting that if a patient has low p140Cap, restoration of this could restore anti-tumor immune responses.

The authors provide a succinct yet comprehensive overview of the role of p140Cap/SRCIN1 as an inhibitor of tumor growth and metastasis and expand on the knowledge of these proteins with

evidence in support of their role affecting intratumoral MDSC infiltration and suppression, which is known to influence tumor progression. Overall the experiments are very well designed and provide convincing data from both in vitro and in vivo assays in support of their findings.

Specifically, they start with a retrospective analysis of TIL infiltration correlated with p140Cap high vs. low expression that demonstrates significant correlation between p140Cap high status and enhanced TIL infiltration in the entire patient population. In addition, analysis of TCGA data demonstrates the Hallmark Inflammatory response gene set is inversely correlated with SRCIN1 expression. This analysis adequately points out the relevance of the following studies in the context of patient data which is important rationale for the subsequent studies.

Next, preclinical studies are outlined to Restoration of p140Cap expression in TuBo and 4T1 tumor models, both with low expression of p140Cap at baseline, demonstrates restoration of p140Cap delays tumor growth and improves metastatic progression. Flow cytometry of tumors growing the p140Cap restored tumors demonstrated higher levels of CD8 T cells, NK cells, M1 macrophages and decreased M2 macrophages. With regard to MDSCs, p140Cap restored tumors had fewer G-MDSCs and no change in M-MDSCs. To determine if there were any changes in suppressive function, an MDSC suppression assay was performed with G-MDSCs isolated from tumors and demonstrate no difference in suppressive activity as compared to control and as compared to splenic G-MDSCs. This is an excellent set of experiments which convincingly lead to the conclusion that p140Cap affects MDSC infiltration but not function.

The authors further interrogate their hypothesis using depletion of G-MDSCs with a mAb against Ly6G which demonstrated a similar effect on tumor growth and metastasis as seen in p140Cap restored tumors and further supports p140Cap exerts a tumor growth inhibitory effect due to its ability to suppress G-MDSC infiltration of the TME.

To investigate mechanisms of action, RNAseq was performed on tumor cells and revealed expression of Csf3 which encodes G-CSF as one of the top transcriptionally downregulated genes in p140Cap restored tumor cells suggesting that p140Cap promotes anti-tumor activity via interference of release of G-CSF. Real time PCR and ELISA for Csf3 and G-CSF confirm decreased levels in p140Cap restored tumor cells. This was confirmed on tumors at the mRNA level.

Authors evaluated G-MDSC levels in the blood, spleen and bone marrow and note a decrease in blood and spleens of p140Cap restored tumor bearing mice. In addition, evaluation of G-MDSCs in the lungs of p140Cap restored tumor bearing mice also had decreased G-MDSCs. Decreased levels of G-CSF was also measured in vivo by ELISA of serum and demonstrates the correlation of p140Cap restoration with circulating levels of G-MDSCs and a functional marker G-CSF. I appreciated the extra level of evaluation of G-MDSCs within a metastatic compartment. An even more convincing study would be to macrodissect lung metastases and repeat these studies to demonstrate how G-MDSC infiltration and function is affected within the different TMEs of the breast vs. metastatic tumor.

The authors next employ tumor mammospheres to enable the detection of G-CSF production by tumor stem cells as compared to bulk tumor cells. This assay was done in 4T1, TuBO and MDA-MB-231 cells with restored p140Cap and demonstrated decreased mammosphere forming efficiency suggestive that p140Cap negatively effects tumor stem cell self renewal. Additional in vivo assays were conducted that demonstrate in p140Cap restored tumor cells there is decreased tumorigenic potential of tumor stem cells. FACS analysis confirmed decreased Sca1+ cells and CD44+/CD24- cells as well. These results were validated in 51 cell lines using the CCLE data base as well as in TCGA database which examined evaluation of a gene set representing self-renewal (Wong Embryonic Stem Cell Core). Lastly, the authors measured G-CSF by ELISA in supernatant of mammosphere cultures and once again demonstrate decreased secretion. Overall, these results show that in all three cell lines, p140Cap overexpression invariably induced a reduction in mammosphere-forming efficiency, establishing a direct correlation between p140Cap-induced shrinkage of the stem cell compartment and impaired G-CSF secretion.

The authors then turn to evaluation of the Wnt/Beta-catenin pathway given that they have previously seen that it is a main interactor of p140Cap. While they reference previous work it

would be nice to have a sentence to summarize the connection prior to these analyses. Authors next adequately investigate un-phosphorylated beta-catenin to demonstrate a decrease in activated beta-catenin but not total levels, in mammospheres derived from p140Cap rescued tumor cells. To prove that the mechanism of p140Cap is control of phosphorylation kinetics they mutate the phosphorylation site and perform numerous in vitro and in vivo studies which show that p140Cap controls the beta-catenin/stem cell/G-CSF functional axis.

The authors end their studies by determination that p140Cap stabilizes Beta-catenin destruction complex, which allows for beta-catenin degradation and decrease in Wnt signaling, which ultimately results in a decrease in tumor-secreted G-CSF (and decreased Cs3 expression), which leads to a decrease in MDSC recruitment. Their proof of principal assay utilizing the IWR-1 compound further proves that this mechanism of action is adequate to affect tumor growth and metastatic progression.

Overall, I think one of the most important findings of this impressive study is the overwhelming data to support the use of p140Cap as a suitable biomarker to identify patients that might be eligible for anti-MDSC targeted therapies or those that might benefit the most from adjuvant treatments to restore an effective anti-tumor response. One of the most challenging aspects of using MDSCs as a biomarker is the difficulty in detection in patients and their plasticity between circulation, tumors and different organs. The ability to measure p140Cap as a surrogate is tremendous and as the authors state, creates new opportunities patient stratification for targeted treatments.

Reviewer #3:

Remarks to the Author:

The authors have shown that p140Cap expression in breast cancer cell lines leads to reduced G-CSF production and reduced PMN-MDSC levels. The result is improved anti-tumor immunity. It also reduces beta-catenin activity. The interplay of these two phenomena could be better defined.

Major Comments:

1. In the Results on please mention the cut-offs for high vs. low p140Cap. What is the TIL infiltration in the very highest lowest p140 Cap segments (top/bottom 10 or 20% for instance?)
2. The phrase "Collectively, these results indicate that 12 p140Cap can modulate the TME immune infiltrate and the anti-tumor inflammatory response in BC 13 patients" is too strong. There is a correlation only at this point. Revised wording would be appropriate.
3. Is it possible to administer PMN-MDSC to tumor bearing mice (i.p. or i.v.) to determine their effect on the growth of TuBo and 4T1 tumors that over-express p140Cap?
4. Is there a dose-response effect on tumor growth with respect to expression level of p140Cap (some over-expressing cell lines have more than others).
5. It was observed that p140Cap expression was inversely correlated with the expression of β Catenin target genes in the TIC (Tumor-Initiating Cell) compartment. Thus, p140Cap direct effects on tumor cell proliferation/stemness could be a factor in reduced growth of p140Cap over-expressing tumors. Is there a sense of the contribution of p140Cap indirect immune effects and direct cancer cell effects on tumor growth? Is there a way to untangle the direct and indirect effects of p140Cap and determine their relative contribution?
6. In 2020, P. Daek et al. published a study in the journal Blood that Catenin-TCF/LEF signaling promotes steady-state and emergency granulopoiesis via G-CSF receptor upregulation. This manuscript should probably be referenced and evaluated in a revised Discussion.

Minor Comments:

1. The word "Remarkably" should be used infrequently and replaced perhaps with "Notably" or even nothing at all.

POINT BY POINT REPLY TO THE REVIEWERS' COMMENTS (reproduced verbatim)

Reviewer #1 - Expertise in breast cancer, tumour initiating cells, b-catenin -(Remarks to the Author):

We would like to thank Reviewer #1, as they provided a number of insightful comments that prompted us to perform a series of additional studies, whose results helped us to craft an improved version of our manuscript. The major points from this Reviewer are reproduced verbatim below and numbered for convenience:

1) This is a continuation of this group's studies of the role of the adaptor protein and putative tumor suppressor p140Cap in cancer primarily in neuroblastoma and more recently breast cancer. This study focuses on its role in TICs, how it might regulate the β -catenin destruction complex and the tumor immune microenvironment. Previous studies have suggested a role for p140CAP through its interaction with p130Cas on Src activation. The authors use two "syngeneic" Balb/c models, the well studied 4T1 model and TuBo cells from MMTV-NeuT mice. Since NeuT is a rat gene this model may not be totally "syngeneic" and this may partially impact their immune profiling.

R. We acknowledge the relevance of the Reviewer's point. In response to this point, we would like to note that the impact of p140Cap on the stromal immune response in the TuBo model was highlighted through the side-by-side analysis of mock vs. p140Cap-infected cells which, as previously shown (Grasso et al., Nat Commun., 2017), express comparatively similar levels of the NeuT rat antigen. Based on this evidence, it is therefore reasonable to argue that the differences in the composition of the immune infiltrate upon xenografting of mock vs. p140Cap TuBo cells are to be ascribed to the intrinsic differences in p140Cap expression, nullifying the possible influence of NeuT rat gene. However, being aware of the relevance of the point raised by this Reviewer, we also leveraged the syngeneic 4T1/Balb/c breast cancer model, which entirely recapitulated the results obtained in the comparison of mock vs. p140Cap TuBo cells, further supporting the notion that the observed effects on the stromal immune infiltrate in the two syngeneic models depend exclusively on the presence or absence of p140Cap.

2) My main concern with this study is the lack of mechanistic insight about how p140Cap regulates the destruction complex and how this leads to an increase in G-CSF. These two observations are the real novelty of this study.

R. We acknowledge the relevance of this point and we thank the Reviewer for their appreciative words on the potential novelty of our findings. Prompted by this Reviewer's comment, we have made a substantial effort to highlight the mechanism through which p140Cap regulates the destruction machinery. In particular, in this revised version of the manuscript we:

1) provide an entirely new set of results based on biochemical (a) and imaging microscopy (b) studies directly in living cells that better define, from a mechanistic standpoint, the role of p140Cap in the context of the destruction complex. These findings converge on the demonstration that p140Cap operates as a stabilizer of the β -Catenin destruction machinery, thereby promoting β -Catenin inactivation in the context of the destruction complex *per se*. In detail, now we show a new set of findings based on:

a) in-depth biochemical characterization of p140Cap as an integral component of the destruction complex, leveraging the BirA biotin/protein ligase-based BioID assay, which is an amenable and highly efficient approach to detect, directly in living cells, proximal protein

interactions in a range of 10 to 20 nm. The sum of these biochemical experiments (illustrated in the new Figure 7 and described in the revised text at page 15, lines 345 – 364) show that, in addition to enter the destruction complex (already argued for by results from the series of reciprocal immunoprecipitation experiments with the different components of the complex included in the previous version of the study), p140Cap overexpression also:

i) increases the interaction among the components of the destruction complex;
ii) causes an increased entrapment of β -Catenin within the complex. Importantly, the increased amount of β -Catenin interacting with p140Cap in these proximity biotinylation experiments appears to be by and large in its inactive form, as testified by its triple-phosphorylated state, which is a post-translational modification executed by the coordinated and sequential action of CK1 and GSK3beta required to prime beta-catenin for subsequent ubiquitination and proteosomal degradation.

Consistently, we also show now that, in the presence of the proteasome inhibitor MG132, significantly exceeding levels of ubiquitinated β -Catenin can be detected in the comparison of mammospheres from p140Cap vs. mock Tubo cells (described at page 15, line 345- 350 of the revised text and shown in the new Figure 7F).

b) extensive immunofluorescence imaging studies in living cells demonstrating that:

i) p140Cap localizes in liquid-liquid phase separation (LLPS) spherical punctate structures, which are spatially confined compartments recently shown to operate as Axin1-dependent docking platforms for the recruitment of the destruction complex components, and where biochemical reactions and signaling events in the Wnt/ β -Catenin pathway take place (Alberti et al., 2019 Cell; Nong et al 2021 JCB; Faux et al, 2008 Oncogene; Schaefer and Peifer, 2019 Dev Cell).

ii) p140Cap expression increases the absolute number of these Axin1 LLPS structures.

iii) p140Cap expression favors Axin1 immobilization in the context of LLPS punctae, as established in FRAP experiments measuring Axin1-GFP fluorescent recovery in TuBo p140Cap-RFP vs. mock transfected cells.

The sum of these imaging studies are described at page 15/16, line 365 - 387 of the revised text and shown in the new Figure 8.

Collectively, these biochemical and imaging studies provide substantial evidence that the mechanistic site of action of p140Cap inhibition over β -Catenin resides in its ability to enter the β -Catenin destruction machinery and to control the dynamics of its formation and stabilization, thereby enhancing β -Catenin inactivation.

2) We also provide a more accurate description of the inextricably intertwined connection between p140Cap ability to restrict the tumor initiating cell (TIC) compartment, that leads to a decreased secretion of G-CSF, in the bulk breast cancer cell population (described in Supplementary S5A), clarifying that the reduced secretion of G-CSF is a direct consequence of the decreased TIC compartment in the bulk population (see page 12, line 290 - 293 of the revised text).

3) The other observations on correlating TILs with patient outcome, G-CSF regulating MDSCs, β -catenin regulating TICs are quite predictable from earlier studies.

R. Related to this point, while we agree with the Reviewer that the prognostic relevance of TILs to breast cancer has already been established, as has been the regulatory role of G-CSF on PMN-MDSCs and the role of β -catenin in TIC regulation, we believe that our study elucidates an entirely novel aspect of the tumor suppressor function of p140Cap, through the identification and functional validation of a previously uncharacterized epistatic β -catenin/TIC/G-CSF/PMN-MDSC axis, which integrates tumor-autonomous and tumor

extrinsic effects, where p140Cap operates as an upstream downregulatory effector of the entire axis. We also submit that another relevant novel contribution of this study is the identification of the mechanism sitting at the apex of the downregulation of this functional axis, which we identified in the ability of p140Cap to act as a stabilizer of the β -catenin destruction machinery. This latter finding also opens new avenues for pharmacological interventions to target the dysfunction of the β -catenin signaling pathways, as formally argued for by the use of the IWR-1 compound in *in vitro* and *in vivo* models (see Figure 9, and page 17 in the Results, and page 20, lines 482 - 488 in the Discussion of this revised version).

4) The actual effect on tumor growth of genetically manipulating p140Cap levels (Fig. 2A) are modest, and the treatment response with IWR-1 (Fig. 8E) is not that impressive.

R. Acknowledged. In response to this Reviewer's point, we would like to note that differences between mock and p140Cap-injected mice are highly significant, and that we observed a consistent behavior in two different preclinical models, namely TuBo and 4T1 tumors, both in term of tumor outgrowth and, even more importantly, in spontaneous lung metastasis (Figure 2A). We also note that these experimental data are consistent with our clinical observation that p140Cap expression correlates in real-life breast cancer patients, including HER2+ patients, with a more favorable prognosis.

As to the effects of IWR-1, as shown in the Figure 9E, we note that treatment with this drug efficiently curbs tumorigenesis in mock TuBo cells, causing a reversion of tumor growth comparable to that observed in p140Cap-overexpressing tumors. We also note that IWR-1 fully mimics the effects observed in p140Cap tumors in terms of inhibition of active β -catenin levels, G-CSF secretion and modifications of the tumor immune infiltrate.

We therefore believe that the sum of these findings makes a cogent point as to the potential of IWR1 to entirely recapitulates the tumor suppressive function of p140Cap.

5) I am not convinced that “basally low p140Cap expression will be a useful biomarker to stratify BC patients”, and that “drugs able to stabilize the β -catenin destruction complex will improve therapeutic response”.

R. We thank the Reviewer for this comment, as it made us realize that the potential translation of p140Cap as a biomarker for patient stratification for targeted therapies, and the therapeutic actionability of the β -catenin signaling pathway, were too succinctly discussed and somehow overstated in the original version of the manuscript.

In this revised version in the results (see page 15, lines 345 – 364) and in the Discussion (page 20 – 21) we now better discuss the potential of p140Cap as a clinical biomarker, when viewed in light of the increasingly recognized role of TICs/cancer stem cells as drivers of tumor progression and metastasis, and the increasing efforts to identify therapeutically actionable pathways to interfere with their function. In addition, based on the set of new findings highlighting the action site of p140Cap inhibition of β -catenin through stabilization of destruction complex, we also better discuss the potential of the β -catenin destruction complex as a therapeutically actionable target to restore an efficient anti-tumor response and inhibit immune tolerance. We submit that this can be achieved using drugs, exemplified by IWR-1, that might inhibit the epistatic β -catenin/TIC/G-CSF/PMN-MDSC axis through the same mechanism of destruction complex stabilization enacted by p140Cap. We believe that the discussion of these points is of the utmost relevance to the naturally occurring breast cancer disease considering that, as shown in the new Supplementary Figure S3 of this revised version (see also our response to minor point 2 from this Reviewer, below), breast tumors with the lowest levels of p140Cap belong to the triple-negative subtype, notably characterized by a more aggressive biology and adverse clinical course. As these tumors also have limited therapeutic options compared to other subtypes, such luminal and HER2+, they might represent an ideal

clinical setting for the use of p140Cap as a biomarker to identify those patients eligible for future clinical studies for the validation of drugs against the β -catenin pathway, some of which present promising therapeutic potential in preclinical studies and clinical trials of some cancer types (reviewed in Jung and Park, *Exp. Mol Med*, 2020).

6) The blood levels of PMN-MDSCs may provide a better and less invasive alternative than measuring p140Cap.

R. Acknowledged. We agree with the Reviewer that, in principle, direct PMN-MDSC measurement in the bloodstream would represent a suitable, liquid biopsy-like, approach to detect the occurrence of a pro-tumorigenic vs. anti-tumorigenic response in a given patient. We note however that, given the variety of alternative mechanisms, other than the dysfunction of the p140Cap/ β -catenin circuitry identified in our study that may converge on altering the relative percentage of resident and circulating PMN-MDSC, an increase in circulating PMN-MDSC might not necessarily reflect a p140Cap/ β -catenin axis alteration. We therefore submit that an accurate assessment of p140Cap status in the primary tumor, which can simply be obtained on routine immunohistochemistry samples, might represent a more suitable “surrogate” proxy for screening patients for possible alteration of their immune response, related to the dysfunction of the epistatic β -catenin/TICs/G-CSF/PMN-MDSCs axis (please, see also point 3 above).

We also note that another major hurdle to the routine implementation of circulating PMN-MDSCs as a liquid biopsy biomarker is the promiscuity and lack of specificity of the most common MDSC cell surface markers, such as CD14, HLA-DR, CD15, PD-L1, among others, which are notably shared with other immune myeloid components (Bronte et al, *Nat Commun*. 2016). This, alongside the well-established MDSC cell plasticity and their ability to fluctuate across different maturation states, does not allow to unequivocally identify MDSCs from other circulating leukocytes. This has likely represented the major obstacle so far to the development of ‘clinical grade’ diagnostic tools for routine use, despite increasingly mounting evidence of the correlation between MDSC accumulation and cancer progression and/or metastatic spreading in different cancer types.

7) Drugs regulating β -catenin have shown little utility to date, and have potential toxicities.

R. We thank the reviewer for this comment, as it concerns a more general aspect related to the actual translation of preclinical research findings to the patient’s bedside. Indeed, if it is true that the Wnt/ β -catenin signaling pathway is increasingly emerging as a promising therapeutic target for cancer intervention in basic research and preclinical studies, the potential toxicity of anti-Wnt/ β -catenin drugs remains a challenge towards their effective translation to the clinical practice, consistent with the well-established role of this pathways in the homeostatic control of developmental programs in several normal tissues. Notwithstanding, as stated above (please, see also point 5) there exists a worldwide increasing effort to identify agonists, antagonists and inhibitory modulators targeting at multiple levels the Wnt/ β -catenin signaling, and some of these compounds present promising therapeutic potential in preclinical studies and clinical trials of some cancer types (reviewed in Jung and Park, *Exp. Mol Med*, 2020).

In this context, while we entirely agree with the Reviewer that anti-Wnt/ β -catenin drugs have a high toxicity potential given their potential to interfere with normal tissue homeostasis, we also believe that the therapeutic index, i.e. the actual therapeutic/beneficial vs. toxic/adverse effects, of these drugs can be considerably improved by accurate patient stratification for the actual dependency of their neoplasms on the Wnt/ β -catenin pathway dysfunction. In this context, it is worth noting that the identification of biomarkers for therapy response prediction, enabling the identification of patients that can safely benefit from de-escalating protocols that minimize the adverse side-effects, while maximizing the therapeutic index, is a general

problem inherent to the development of new targeted therapies. Notably, this is still an unmet need even in the case of a variety of molecularly targeted treatments already in clinical use. Therefore, as drugs directed against the Wnt/ β -catenin signaling pathway make no exception in this context, we submit that p140Cap might clinically behave as one such biomarker for patient stratification, based on i) our preclinical evidence that it acts as a stabilizer of the β -catenin destruction machinery and ii) our proof of evidence of the potential therapeutic actionability of this mechanism using the IWR-1 compound.

This Reviewer had also a number of minor concerns that are reproduced verbatim below and numbered for convenience:

1) Page 3, Line 3 – Update estimated cases for 2022.

R. Agree. According to this Reviewer's suggestion, we have updated to 2022 the numbers related to the breast cancer disease, and inserted a new appropriate reference (see page 3, line 39 - 40).

2) Page 6, Line 6 – What about triple negative breast cancers? TNBCs also have a high risk of metastatic recurrence. What are the levels of p140Cap in TNBCs?

R. Acknowledged. To address this relevant Reviewer's point, we have i) analyzed the distribution of p140Cap protein levels by immunohistochemistry across the different molecular subtypes in our cohort (the same used for TIL analysis in the original version) and ii) performed a meta-analysis of p140Cap/*SCRCINI*, both at the protein and transcript level, in a publicly available cohort of 1095 breast cancer patients from the TCGA dataset and in the IEO cohort. The sum of these analyses, shown in the new Supplementary Fig. S3A, B and discussed in the revised text at page 6 line 122 - 132, show that triple-negative breast cancers feature significantly lower p140Cap levels compared to both luminal and HER2+ breast cancer subtypes, a finding consistent with the more adverse clinico-prognostic behavior of triple-negative tumors compared to the other molecular subtypes.

3) Page 6, Line 14- Authors need to do additional experiments to identify TILs, either by IHC staining of CD3+ and CD20+ or Immunofluorescence or multiplex IF.

R. Acknowledged. Following this Reviewer's indication, we have performed immunofluorescence analysis with the indicated cell surface markers to characterize the lymphocyte infiltration in a set of breast tumors with proficient p140Cap expression. The results are illustrated in the new Supplementary Figure S2 (at page 6, line 115 – 118) and show that the immune infiltrate surrounding p140Cap-expressing tumors comprises the presence of both CD3+/T-lymphocytes and CD20+/B-lymphocytes, indicative of a proficient immune response in the tumor microenvironment.

4) Page 7, Lines 2 – If a correlation exists between TILs and favorable disease outcomes in HER2+ patients, then what are the levels of p140Cap in these patients (aside from correlation analysis)? Have these patients received neo-adjuvant therapy? Why are they infiltrated by lymphocytes?

R. Agree. Prompted by this relevant Reviewer's comment, we have carefully analyzed the distribution of p140Cap as a function of TIL-positivity in HER2-amplified breast cancer patients. Furthermore, to have a comprehensive view of the association between p140Cap and TILs in the entire breast cancer disease, we have also extended this analysis to the other molecular subtypes. Results from these analyses are shown in the Figure 1 and in new Supplementary Fig. S1, S3 and discussed in the revised text at page 5 line 106 to page 6 line 132. These results clearly show that, in HER2+ breast tumors, a p140Cap-positive status

associates with a significantly higher (OR=2.13) to feature positive TIL infiltration, with the percentage of TIL-positive patients raising from less than 30% in the group of p140Cap-low patients to more than 70% in the group of p140Cap-positive patients. We note that the odd ratio (OR=2.13) relative to these analyses in HER2+ patients was not statistically significant ($p=0.30$), most likely due to the very limited size of HER2+ patients ($n=64$) available for this subgroup analysis in our cohort. However, a very similar and statistically significant behavior (OR=2.23, $p=0.04$) could be observed in the cohort of HER2-negative ($n=300$) patients, with TIL-positive patients raising from less than 20% to more than 80% in p140Cap-low vs. p140Cap-high patients, as well as in the entire cohort ($n=364$, OR=2.25, $p=0.02$). Collectively, these results clearly point to a direct association between p140Cap expression and increased rate of TIL infiltration in the stromal microenvironment.

Concerning the Reviewer's question as to whether these HER2+ patients had been subjected to neoadjuvant treatments, we are aware that this point is of the utmost relevance to the well-established correlation between presence of TILs and higher rate of pathological complete response to neoadjuvant anti-HER2 treatments (Denkert C, von Minckwitz G, Brase JC, et al., 2015, J. Clin. Oncol.; Salgado R, Denkert C, Campbell C, et al., 2015, JAMA Oncol.). Regretfully, we note that our study cohort comprises a longitudinal retrospective series of adjuvant patients only, extracted consecutively from years 1997-2000 to have a long follow-up (please, see also Materials and Methods). As in these years anti-HER2 therapies were not yet available in the routine clinical practice, we are not in a position to address this relevant Reviewer's question in our analyses.

In their last question, the Reviewer asks why these HER2+ patients are infiltrated by lymphocytes. Related to this point, we note that, as stated above, the presence of TILs in HER2+ breast tumors is a frequent occurrence that is now also being increasingly considered for its potential therapy prediction and prognostic value. Consistently, we show that high TIL infiltration correlates with a significantly more favorable prognosis in HER2+ (see Figure S1), and that HER2+ patients with high p140Cap expression have a higher propensity to show TIL positivity. Therefore, p140Cap appears to represent a key determinant of stromal tumor infiltration to instruct an efficient anti-tumor immune response in these patients. We also note that the *SRCINI*/p140Cap gene is frequently co-amplified (from 50 to 60%) with the *HER2* gene (Grasso et al. Nat Commun, 2017), which is likely the explanation for the frequent TIL infiltration observed in HER2+ breast tumors.

5) Page 7, Line 7 – Is the TCGA dataset of 1095 transcripts from HER2+ patients, TNBC, ER+? If the authors have these data they can study the transcript levels of SRCINI within each subtype.

R. Agree. This point is related to another major points raised by this Reviewer (please, see point 2) and we therefore refer the Reviewer to our detailed reply above, where we describe the distribution of p140Cap/*SCRCINI*, both at the protein and transcript level, across the different molecular subtypes of breast cancer obtained from the meta-analysis of the TGCA dataset, alongside the results obtained from immunohistochemical analysis of p140Cap expression in the different molecular subtypes in our cohort from the European Institute of Oncology (Supplementary Figure S3A, B).

6) Page 8 – Figure 2A, the authors should display images of H&E stained lung tissues to help confirm these findings.

R. Agree. According to the Reviewer's indication, we have now included in Supplementary Figure 2, panel C-D, a set of representative H&E images of lung tissues from mice injected

with mock or p140Cap TuBo, or 4T1 cells, where each figure shows the metastatic area (see also Materials and Methods).

7) Page 8 – *What markers were utilized to distinguish between M1 and M2 like macrophages?*

R. Agree. The configuration CD11b+/F480+/MHC-II+ and CD11b+/F480+/CD206+ were used to prospectively purify by FACS the M1 and M2 macrophage population, respectively. This information was already present in Materials and Methods of the previous version.

8) Page 13 – *Why use MDA-MB-231 since the manuscript was focused on HER2+ BCs?*

R. In response to this Reviewer's point, we would like to note that the newly identified tumor suppressive function of p140Cap in promoting an anti- vs. pro-tumoral response, through its inhibitory control over a functional β -catenin/TIC/G-CSF/PMN-MDSCs axis, was found to be relevant to both the syngeneic HER2+ TuBo and the syngeneic triple-negative 4T1 model, suggesting that this p140Cap function extends to breast cancer subtypes beyond the HER2+ subtype. Further supporting this view, in this revised version we show that:

- a) triple-negative breast cancer show significantly reduced p140Cap levels compared to all the other subtypes (please, see also our response to point 2 and 5 above);
- b) the association between high p140Cap status and TIL infiltration is invariably relevant to all the different molecular subtypes (see also our response to point 4 above).

Reviewer #2 - Expertise in cancer immunology, MDSCs (Remarks to the Author):

The authors demonstrate using transcriptomics and preclinical mouse models that p140Cap expression correlates with enhanced presence of TIL and counteraction of MDSCs and can be considered a novel tumor suppressor. Specifically decreased G-CSF via p140Caps stabilizes of beta-catenin in the tumor initiating cell compartment/ tumor stem cells, which leads to prevention of MDSC infiltration and as a result decreased suppression of the TME. The authors also propose p140Cap could be a biomarker to help stratify tailored therapy. Specifically suggesting that if a patient has low p140Cap, restoration of this could restore anti-tumor immune responses.

The authors provide a succinct yet comprehensive overview of the role of p140Cap/SRCIN1 as an inhibitor of tumor growth and metastasis and expand on the knowledge of these proteins with evidence in support of their role affecting intratumoral MDSC infiltration and suppression, which is known to influence tumor progression. Overall the experiments are very well designed and provide convincing data from both in vitro and in vivo assays in support of their findings.

Specifically, they start with a retrospective analysis of TIL infiltration correlated with p140Cap high vs. low expression that demonstrates significant correlation between p140Cap high status and enhanced TIL infiltration in the entire patient population. In addition, analysis of TCGA data demonstrates the Hallmark Inflammatory response gene set is inversely correlated with SRCIN1 expression. This analysis adequately points out the relevance of the following studies in the context of patient data which is important rationale for the subsequent studies.

Next, preclinical studies are outlined to Restoration of p140Cap expression in TuBo and 4T1 tumor models, both with low expression of p140Cap at baseline, demonstrates restoration of p140Cap delays tumor growth and improves metastatic progression. Flow cytometry of tumors growing the p140Cap restored tumors demonstrated higher levels of CD8 T cells, NK cells, M1 macrophages and decreased M2 macrophages. With regard to MDSCs, p140Cap restored tumors had fewer G-MDSCs and no change in M-MDSCs. To determine if there were any

changes in suppressive function, an MDSC suppression assay was performed with G-MDSCs isolated from tumors and demonstrate no difference in suppressive activity as compared to control and as compared to splenic G-MDSCs. This is an excellent set of experiments which convincingly lead to the conclusion that p140Cap affects MDSC infiltration but not function.

The authors further interrogate their hypothesis using depletion of G-MDSCs with a mAb against Ly6G which demonstrated a similar effect on tumor growth and metastasis as seen in p140Cap restored tumors and further supports p140Cap exerts a tumor growth inhibitory effect due to its ability to suppress G-MDSC infiltration of the TME.

To investigate mechanisms of action, RNAseq was performed on tumors cells and revealed expression of Csf3 which encodes G-CSF as one of the top transcriptionally downregulated genes in p140Cap restored tumor cells suggesting that p140Cap promotes anti-tumor activity via interference of release of G-CSF. Real time PCR and ELISA for Csf3 and G-CSF confirm decreased levels in p140Cap restored tumor cells. This was confirmed on tumors at the mRNA level.

Authors evaluated G-MDSC levels in the blood, spleen and bone marrow and note a decrease in blood and spleens of p140Cap restored tumor bearing mice. In addition, evaluation of G-MDSCs in the lungs of p140Cap restored tumor bearing mice also had decreased G-MDSCs. Decreased levels of G-CSF was also measured in vivo by ELISA of serum and demonstrates the correlation of p140Cap restoration with circulating levels of G-MDSCs and a functional marker G-CSF. I appreciated the extra level of evaluation of G-MDSCs within a metastatic compartment.

An even more convincing study would be to macrodissect lung metastases and repeat these studies to demonstrate how G-MDSC infiltration and function is affected within the different TMEs of the breast vs. metastatic tumor.

R. We thank Reviewer #2 for the appreciative words and for highlighting the novelty of our findings linking the tumor suppressor function of p140Cap to its intrinsic ability to instruct an anti-tumorigenic vs. a pro-tumorigenic immune response in the tumor microenvironment, and for pointing out the relevance of our study to real-life breast cancer patients.

Among their comments, this Reviewer also had an insightful idea, i.e. to compare the qualitative/quantitative variations in the TME immune infiltrate between the primary tumor vs. the lung metastatic site from mock- vs. p140Cap-tumor bearing mice. We acknowledge that this would represent a nice and elegant complement to the bulk of our data obtained by prospective FACS sorting (depicted in Fig. 4B).

We note however that, albeit feasible in principle, these analyses would require complex multiplexed spatial in situ immunofluorescence studies, given the promiscuity and lack of specificity of surface biomarkers to unequivocally distinguish MDSCs vs. neutrophils. We also note that our FACS studies showing higher frequency of MDSC lung infiltration in mock vs. p140Cap 4T1 tumor-bearing mice were performed as early as 14 days post-injection when, most likely, a fully formed metastasis had not yet developed. We chose this very early time-point for our FACS analyses with the aim to explore the possible effects of p140Cap overexpression on the formation of an early pre-metastatic niche by MDSCs. Instead, to tackle the idea of this Reviewer, we should have allowed mock- and p140cap 4T1 tumor-bearing mice to develop fully formed metastasis and analyze them by multiplexed immunofluorescence analysis, in comparison with the primary tumor, for their respective composition of the TME immune infiltrate. We believe that such an endeavor, which will necessarily take considerable

time to be performed in a systematic way on a sizable number of samples to achieve statistically significant results, will constitute an effort *per se* and will hopefully represent the object of a future publication.

The authors next employ tumor mammospheres to enable the detection of G-CSF production by tumor stem cells as compared to bulk tumor cells. This assay was done in 4T1, TuBO and MDA-MB-231 cells with restored p140Cap and demonstrated decreased mammosphere forming efficiency suggestive that p140Cap negatively effects tumor stem cell self-renewal. Additional in vivo assays were conducted that demonstrate in p140Cap restored tumor cells there is decreased tumorigenic potential of tumor stem cells. FACS analysis confirmed decreased Sca1+ cells and CD44+/CD24- cells as well. These results were validated in 51 cell lines using the CCLE data base as well as in TCGA database which examined evaluation of a gene set representing self-renewal (Wong Embryonic Stem Cell Core). Lastly, the authors measured G-MSCF by ELISA in supernatant of mammosphere cultures and once again demonstrate decreased secretion. Overall, these results show that in all three cell lines, p140Cap overexpression invariably induced a reduction in mammosphere-forming efficiency, establishing a direct correlation between p140Cap-induced shrinkage of the stem cell compartment and impaired G-CSF secretion.

The authors then turn to evaluation of the Wnt/Beta-catenin pathway given that they have previously seen that it is a main interactor of p140Cap. While they reference previous work it would be nice to have a sentence to summarize the connection prior to these analyses (see revised version in Discussion at page, 18, line 418-423). Authors next adequately investigate un-phosphorylated beta-catenin to demonstrate a decrease in activated beta-catenin but not total levels, in mammospheres derived from p140Cap rescued tumor cells. To prove that the mechanism of p140Cap is control of phosphorylation kinetics they mutate the phosphorylation site and perform numerous in vitro and in vivo studies which show that p140Cap controls the beta-catenin/stem cell/G-CSF functional axis.

The authors end their studies by determination that p140Cap stabilizes Beta-catenin destruction complex, which allows for beta-catenin degradation and decrease in Wnt signaling, which ultimately results in a decrease in tumor-secreted G-CSF (and decreased Csf3 expression), which leads to a decrease in MDSC recruitment. Their proof of principal assay utilizing the IWR-1 compound further proves that this mechanism of action is adequate to affect tumor growth and metastatic progression.

Overall, I think one of the most important findings of this impressive study is the overwhelming data to support the use of p140Cap as a suitable biomarker to identify patients that might be eligible for anti-MDSC targeted therapies or those that might benefit the most from adjuvant treatments to restore an effective anti-tumor response. One of the most challenging aspects of using MDSCs as a biomarker is the difficulty in detection in patients and their plasticity between circulation, tumors and different organs. The ability to measure p140Cap as a surrogate is tremendous and as the authors state, creates new opportunities patient stratification for targeted treatments.

R. Once again, we are most grateful to Reviewer #2 for their full appreciation of the amount of work performed in this study and, even more, for their appreciation of the potential translatability of these findings to real-world breast cancer patients. In this last part of their comments, this Reviewer made us note that, in support of the rationale and background of our

study, we too succinctly dealt with the previously observed regulatory role of p140Cap over the Wnt/ β -catenin pathway. Following the Reviewer's suggestion, we have included a sentence to expand the background of our study **in the Introduction at page 4 line 70 - 73**.

Reviewer #3 - Expertise in breast cancer, MDSCs - (Remarks to the Author):

The authors have shown that p140Cap expression in breast cancer cell lines leads to reduced G-CSF production and reduced PMN-MDSC levels. The result is improved anti-tumor immunity. It also reduces beta-catenin activity. The interplay of these two phenomena could be better defined.

R. Agree. We thank the Reviewer for this comment, as it prompted us to provide a more careful explanation of the major finding of this study, i.e. the identification of an epistatic β -catenin/TIC/G-CSF/PMN-MDSC axis in which tumor-autonomous and tumor-extrinsic suppressive effects of p140Cap are inextricably integrated towards the promotion of an efficient anti-tumor immune response, while preventing a tumor conducive immune tolerance. The intertwined interdependency between cell-autonomous effects of p140Cap in downregulating TIC self-renewal and tumorigenicity and secretion of G-CSF, on the one hand, and local and systemic effects due to reduced G-CSF secretion and MDSC mobilization, on the other hand, is now extensively discussed at several points of this revised version of the manuscript **in the Results (Figure 4, page 10-11, line 222 – 248; Figure 5 (G-H) page 12, line 287 - 293) and Discussion (page 18 -19, line 438 -455) sections**.

This Reviewer had a number of major points (the different points are reproduced verbatim and numbered according to the original reviewer's points for convenience):

1. In the Results on please mention the cut-offs for high vs. low p140Cap. What is the TIL infiltration in the very highest lowest p140 Cap segments (top/bottom 10 or 20% for instance?)

R. Agree. While in our previous version we only referenced previous work (Grasso et al. Nat Commun., 2017), we have now reported in the Results section that patients were assigned an intensity score from 0 to 3 (0, 0.5, negative; 1+, weak positive; 2+, moderate positive; and 3+, strong positive), and stratified as p140Cap^{LOW} (<1) or p140Cap^{HIGH} (≥ 1). This information has also been reported in Materials and Methods of this revised version, AT PAGE 22, LINE 516 - 517.

To address the Reviewer's point, we analyzed TIL infiltration in the patients with the highest (2+, 3+) or lowest (0, 0.5+) p140Cap intensity scores, excluding patients with intermediate/weak positivity (1+), obtaining groups of comparable size for meaningful statistical evaluation. As expected, by Spearman analysis, the analysis of patients with the highest vs. lowest p140Cap levels resulted in an even more stringent correlation between a positive TIL status and high p140Cap expression (OR=5.30, $p=0.00055$), compared to the results including intermediate patients (see **Figure 1 of this revised version**). This difference between the highest and the lowest p140Cap patients further increased considering only the luminal HER2-negative patients (OR=8.26, $p=0.002$), likely due to the exclusion of the confounding effect of HER2+ patients who frequently bear concomitant p140Cap/HER2 amplification. We are reporting these results for this Reviewer's perusal only below, as we believe that they do not add much further information to the bulk of findings obtained considering all the patients with high, low and intermediate p140Cap scores. However, we are willing to include them in an additional supplementary figure in this revised version, if the Reviewer deems so.

ENTIRE COHORT				HER2-negative patients			
	TIL-neg	TIL-pos	Total		TIL-neg	TIL-pos	Total
p140 low (<1)	15 (18.5%)	66 (81.5%)	81	p140 low (<1)	13 (21.3%)	48 (78.7%)	61
P140 high (>2)	3 (4.1%)	70 (95.9%)	73	P140 high (>2)	2 (3.2%)	61 (96.9%)	63
Total	18	136	154	Total	15	109	124

OR = 5.30 (1.47; 19.16)
p = 0.0055

OR = 8.26 (1.78; 38.38)
p = 0.002

2. The phrase “Collectively, these results indicate that p140Cap can modulate the TME immune infiltrate and the anti-tumor inflammatory response in BC patients” is too strong. There is a correlation only at this point. Revised wording would be appropriate.

R. Agree. We acknowledge the relevance of this comment and we have rephrased the original sentence according to this Reviewer’s suggestion in the revised discussion.

3. Is it possible to administer PMN-MDSC to tumor bearing mice (i.p. or i.v.) to determine their effect on the growth of TuBo and 4T1 tumors that over-express p140Cap?

R. We acknowledge that the experiment of PMN-MDSC adoptive transferring suggested by this Reviewer would be an elegant approach to directly implicate this subclass of immune cells in the pro-tumorigenic vs. anti-tumorigenic response observed in the absence or presence of p140Cap, respectively. However, as previously described for Reviewer 2, the link between tumor-derived G-CSF-dependent increase of PMN-MDSCs, tumor progression and metastatic spreading has been already unveiled and confirmed in many tumor preclinical models, breast included (Chafe et al., Cancer Res, 2015; Kowanzetz et al., PNAS, 2010), and breast cancer patients (Lawicki et al., Adv Med Sci, 2013). Higher G-CSF levels supports hematopoietic rewiring and PMN-MDSC recruitment in both primary tumors, peripheral immune districts and metastatic niche. We confirmed these findings in our study and unveiled a precedent-unrecognized role for p140Cap in regulating G-CSF secretion by tumor cells, tumor proliferation and metastatic spreading. Moreover, we mechanistically challenged p140Cap-dependent, control of tumor progression by bypassing p140Cap-mediated regulation of β -Catenin destruction complex (C.A. β -Catenin) or by stabilizing it (IWR-1 treatment). According with our hypothesis the two antithetical approaches hindered or replaced p140Cap-dependent tumor restriction (Figure 6K, Figure 9E) by controlling G-CSF levels (Figure 6I, 9G).

Notwithstanding, we believe that we have provided a strong proof-of-concept in support of the idea put forward by this Reviewer in our experiments leveraging the intraperitoneal administration of a monoclonal antibody (Mab), 1A8, specifically raised against the Ly6G surface marker and capable of selectively suppressing the PMN-MDSC sub population. As a matter of fact, targeting PMN-MDSCs selectively curbed tumorigenesis in mock-infected TuBo tumor-bearing mice, with a tumor growth inhibitory efficacy comparable to that observed with p140Cap overexpression. These results were included in Supplementary Figure S5E, F, but we are willing to move them to the main text, if the Reviewer deems so.

4. Is there a dose-response effect on tumor growth with respect to expression level of p140Cap (some over-expressing cell lines have more than others).

R. We acknowledge the relevance of this point. In response to this point, we would like to note that, despite intrinsic differences in their total levels of exogenously expressed p140Cap levels,

p140Cap-TuBo and p140Cap-4T1 cells show a similar rate of tumor growth inhibition compared to their respective mock counterparts. In keeping with this, in either p140Cap-expressing cell line, we observed a very similar behavior of the β -catenin/TIC/G-CSF/PMN-MDSCs axis. Related to this point, we can only speculate that, despite their different baseline expression of p140Cap, in either cell line, we were able to achieve a p140Cap overexpression beyond the required threshold to fully display its tumor suppressor function.

5. It was observed that p140Cap expression was inversely correlated with the expression of β Catenin target genes in the TIC (Tumor-Initiating Cell) compartment. Thus, p140Cap direct effects on tumor cell proliferation/stemness could be a factor in reduced growth of p140Cap over-expressing tumors. Is there a sense of the contribution of p140Cap indirect immune effects and direct cancer cell effects on tumor growth? Is there a way to untangle the direct and indirect effects of p140Cap and determine their relative contribution?

R. We acknowledge the utmost relevance of this point to the tumor suppressor function of p140Cap. Of note, in our previous studies, we characterized p140Cap as a potent cell-autonomous tumor suppressor, based on its ability to inhibit tumor growth in xenografting experiments performed in nude mice with the TuBo model (Grasso et al., Nat Commun, 2017). In the present study, using ‘less artifactual’ and more preclinically relevant models, i.e. syngeneic murine tumor transplantation models, we highlight the existence of an epistatic p140Cap/ β -catenin/TIC/G-CSF/MDSC regulatory axis that influences the qualitative/quantitative composition of the stromal immune infiltrate, balancing a pro-tumorigenic vs. anti-tumorigenic immune response.

In this axis, the tumor-autonomous inhibition of p140Cap over β -Catenin signaling results in restriction of the TIC self-renewal ability associated with tumor growth impairment, which is an expected outcome consequent to the cell-autonomous inhibition the TIC/stemness tumorigenic potential. However, these tumor-autonomous events appear to be inextricably linked to a series of tumor-extrinsic events occurring both locally, i.e. at the site of the primary tumor, and at distant sites, i.e. at the level of the peripheral circulation, the spleen and the lung tissue. Apparently, these depend on the impaired production of the inflammatory cytokine, G-CSF, which controls local and systemic dynamics of MDSC mobilization and tissue infiltration and the consequent establishment of an impaired immune tumor surveillance that, in turn, contributes to tumor growth.

We therefore believe that providing an answer to the relevant Reviewer’s question is extremely difficult, if not virtually impossible, considering the well-established function of inflammatory cytokines in integrating local tissue responses with distant effects at the systemic level. For the time being, we can only provide a tentative answer extrapolating from the set of results obtained in experiments in which MDSCs were selectively targeted using an inhibitory monoclonal antibody: these experiments revealed that such a treatment is sufficient to fully recapitulate the growth inhibitory effects observed with p140cap overexpression, in the syngeneic TuBo model, arguing for the intrinsic difficulty to deconvolute the bidirectional cross-talk between tumor-autonomous and -extrinsic dynamics in sustaining tumorigenesis in p140Cap low breast tumors. Prompted by this Reviewer’s question, we now extensively discuss this point at several points of the revised version of the manuscript.

6. In 2020. P. Daek et al. published a study in the journal Blood that Catenin-TCF/LEF signaling promotes steady-state and emergency granulopoiesis via G-CSF receptor upregulation. This manuscript should probably be referenced and evaluated in a revised Discussion.

R. Agree. Following the Reviewer’s suggestion, we have included this reference in the Discussion section (see page 19, line 459).

Minor Comments:

1. The word “Remarkably” should be used infrequently and replaced perhaps with “Notably” or even nothing at all.

R. Acknowledged. We have revised the text according to the Reviewer’s suggestion.

Reviewers' Comments:

Reviewer #1:

Remarks to the Author:

The authors have responded to the majority of the concerns of the reviewers and the new data added strengthens and improves this study. This is a nice contribution. One minor typo on line 82 G-CSF.

Reviewer #2:

Remarks to the Author:

The authors have very nicely addressed my concerns and I believe the additional results included in the manuscript strengthen the work significantly and are worthy of publication so that others can learn from this important work. Explanations regarding some of my requests for experimental additions being beyond the scope of this publication are acceptable and I appreciate the consideration of these suggestions.

Reviewer #3:

Remarks to the Author:

Accept

Point by Point response to the reviewers' comment

REVIEWERS' COMMENTS

Reviewer #1 (Remarks to the Author):

The authors have responded to the majority of the concerns of the reviewers and the new data added strengthens and improves this study. This is a nice contribution. One minor typo on line 82 G-CSF.

We thank the reviewer for her/his kind comment.

Reviewer #2 (Remarks to the Author):

The authors have very nicely addressed my concerns and I believe the additional results included in the manuscript strengthen the work significantly and are worthy of publication so that others can learn from this important work. Explanations regarding some of my requests for experimental additions being beyond the scope of this publication are acceptable and I appreciate the consideration of these suggestions.

We thank the reviewer for her/his kind comment.

Reviewer #3 (Remarks to the Author):

Accept

We thank the reviewer for her/his positive comment.